# FitVid: High-Capacity Pixel-Level Video Prediction

## Abstract

An agent that is capable of predicting what happens next can perform a variety of tasks through planning with no additional training. Furthermore, such an agent can internally represent the complex dynamics of the real-world and therefore can acquire a representation useful for many other visual perception tasks. This makes predicting the future frames of a video, conditioned on the observed past and potentially future actions, an interesting task which remains exceptionally challenging despite many recent advances. Existing video prediction models have shown promising results on simple narrow benchmarks but they generate low quality predictions on real-life datasets with more complicated dynamics or broader domain. There is a growing body of evidence that underfitting on the training data is one of the primary causes for the low quality predictions. In this paper, we argue that the inefficient use of parameters in the current video models is the main reason for underfitting. Therefore, we introduce a new architecture, named FitVid, which is capable of fitting the common benchmarks so well that it begins to suffer from overfitting – while having similar parameter count as the current state-of-the-art models. We analyze the consequences of overfitting, illustrating how it can produce unexpected outcomes such as generating high quality output by repeating the training data, and how it can be mitigated using existing image augmentation techniques. As a result, FitVid outperforms the current state-of-the-art models across four different video prediction benchmarks on four different metrics.

## 1 Introduction

Predicting what happens next is a cornerstone of intelligence and one of the key capabilities of humans, which we heavily rely on to make decisions in everyday life (Bubic et al., 2010). This capability enables us to anticipate future events and plan ahead to perform temporally extended tasks. While the machine learning literature has studied a wide range of prediction problems, one of the most direct challenges is to predict raw sensory inputs. In particular, prediction of future *visual* inputs conditioned on a context of past observations – i.e., pixel-level video prediction – encapsulates the challenges of visual perception, modeling of physical events, and reasoning about uncertain behaviors. Video prediction can be formulated as a self-supervised problem, enabling us to use a substantial amount of unlabeled data to provide autonomous systems with powerful predictive capabilities as well as learning rich representations for downstream tasks. Already, video models have been successfully deployed in applications such as robotics (Finn & Levine, 2017; Zhang et al., 2019), simulation (Kim et al., 2021; 2020), compression (Duan et al., 2020; Sulun & Tekalp, 2021) and video synthesis from a single frame (Endo et al., 2019; Nam et al., 2019).

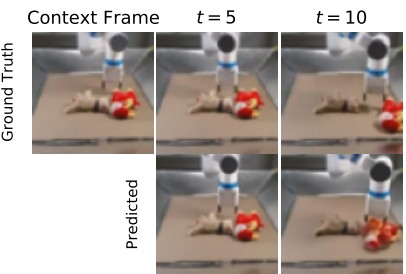

Figure 1: FitVid predicts high quality images of the future given the first few frames. Here, the visual detail on the pushed object is preserved effectively, and even the shadow of the robot's arm is predicted correctly. In this example, FitVid is trained on 15M video frames of 7 diffedrent robots from RoboNet (Dasari et al., 2019).

Despite recent advances in generative models in many domains, such as images (Karras et al., 2017; Vahdat & Kautz, 2020) and text (Devlin et al., 2018; Brown et al., 2020), video prediction is still considered to be challenging. Current state-of-the-art methods are limited to low-resolution videos (typically $64\times64$ (Villegas et al., 2019) and a maximum of $256\times256$ (Kim et al., 2021)), usually in a narrow domain, such as a single human walking Ionescu et al. (2014), or a robotic arm pushing objects in a stationary setting Ebert et al. (2017). Even then the quality of predicted frames tends to drop substantially after as little as 10 seconds into the future (Saxena et al., 2021).

A growing body of evidence suggests that **underfitting** is one of the primary reasons for low quality predictions. For example, Villegas et al. (2019) demonstrate how scaling the model, by adding more parameters, can substantially improve the prediction quality. Similarly, Castrejon et al. (2019) argue that blurry video prediction of variational methods is a sign of underfitting, exhibiting an improved test and train performance as the network capacity increases. Wu et al. (2021a) also observed monotonic improvement as the number of modules in a hierarchical architecture increases. While scaling up models is common to address underfitting, it comes at the cost of more computation, memory, and integration risks (Wolf et al., 2019) as well as more complicated training regimes (Wu et al., 2021a).

In this paper we take a step back and address underfitting by instead finding an architecture that uses its parameters more efficiently. We propose FitVid, a model that – with the same parameter count as current state-of-the-art models – can fit the current video prediction benchmarks successfully. In fact, FitVid fits these benchmarks so well that in many cases we find it actually begins to suffer from *overfitting*. To the best of our knowledge, this is the first time a video model reports *substantial* overfitting on theses benchmarks. Importantly, we also find that simple existing image augmentation techniques can mitigate this overfitting, leading to models that can both fit the training set and generalize well to held-out videos.

As a result, FitVid achieves state-of-the-art on four challenging video datasets across a wide range of metrics. Furthermore, while each of the components of our proposed model are not novel on their own, we find that FitVid can be trained a significantly simpler training recipe. Prior works on video prediction, particularly those that make use of variational methods to provide for stochasticity, typically require a number of delicate design decisions to train successfully: curriculum training (Oh et al., 2015; Lee et al., 2018; Cenzato et al., 2019), a learned prior (Denton & Fergus, 2018; Castrejon et al., 2019) and annealing of the weight on the VAE KL-divergence penalty (Babaeizadeh et al., 2018). In contrast, we show that our method actually fits the training data well without any such components, training directly via optimizing the evidence lower bound with minimal hyperparameters.

## 2 RELATED WORK

Video prediction (Ranzato et al., 2014; Srivastava et al., 2015) has been formulated in different ways such as generating videos from a single image (Endo et al., 2019; Nam et al., 2019; Dorkenwald et al., 2021; Yang et al., 2021; Hu et al., 2021) or no image (Vondrick et al., 2016; Saito & Saito, 2018; Clark et al., 2019), text to video generation (Wu et al., 2021b), video-to-video translation (Wang et al., 2019a; 2018) and data-driven simulation (Kaiser et al., 2019; Kim et al., 2021; 2020). In this paper, our focus is on conditional video prediction, which is to predict the future frames of a video conditioned on a few initial context frames and possibly the future actions of the agents (Finn et al., 2016a; Babaeizadeh et al., 2018). Conditional video prediction has a number of applications, including model-based reinforcement learning from pixels (Hafner et al., 2018; 2019; Kaiser et al., 2019; Rafailov et al., 2020) and robotics (Boots et al., 2014; Kalchbrenner et al., 2017a; Ebert et al., 2017; 2018a;b; Xie et al., 2019; Paxton et al., 2019; Nair & Finn, 2020; Nair et al., 2020).

Initially, video prediction was tackled using deterministic models (Walker et al., 2015; Jia et al., 2016; Xue et al., 2016; Walker et al., 2016; Liang et al., 2017; Byravan & Fox, 2017; Vondrick & Torralba, 2017; Van Amersfoort et al., 2017; Liu et al., 2017; Chen et al., 2017; Lu et al., 2017). Later on, given the common randomness and partial observability in the real-life situations, various stochastic models were proposed to capture the stochasticity of the future. Generative adversarial networks (GANs) (Goodfellow et al., 2014) are used for video predictions (Mathieu et al., 2015; Lee et al., 2018; Clark et al., 2019; Luc et al., 2020; Hong et al., 2021) as sell as autoregressive models (Kalchbrenner et al., 2017b; Reed et al., 2017; Weissenborn et al., 2019). Flow based generative models (Dinh et al., 2014; 2016) are also adopted for video prediction (Kumar et al., 2019).

In this paper we focus on Variational Auto-Encoders (VAEs) (Kingma & Welling, 2014) which are widely used for conditional video prediction (Shu et al., 2016; Babaeizadeh et al., 2018; Denton & Fergus, 2018; Wichers et al., 2018; Villegas et al., 2019; Franceschi et al., 2020; Castrejon et al., 2019; Yan et al., 2021; Rakhimov et al., 2020; Lee et al., 2021; Wu et al., 2021a; Walker et al., 2021). Recently, it has been demonstrated that underfitting on training data plays a major role in low quality blurry predictions in VAE based models. Villegas et al. (2019); Castrejon et al. (2019); Wu et al. (2021a) all reported improved prediction quality as the network capacity increases. In this paper, we are interested in addressing underfitting *without* increasing the capacity of the current models, and instead, find a more expressive architecture which uses its capacity more efficiently.

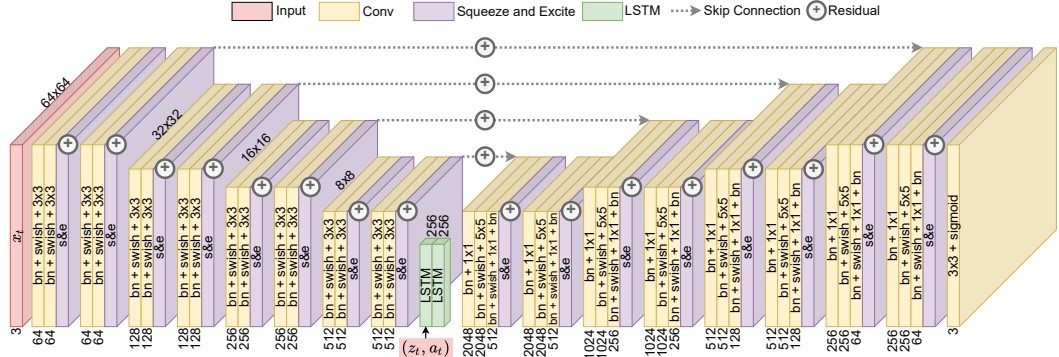

Figure 2: The FitVid architecture. In the figure, **(bn)** is batch-normalization (Ioffe & Szegedy, 2015), **(swish)** is the activation (Ramachandran et al., 2017), **(s&e)** is Squeeze and Excite (Hu et al., 2018) and $(\mathbf{N}\times\mathbf{N})$ is a convolutional layer with kernel size of $\mathbf{N}\times\mathbf{N}$. The strides are always one, except when down-sampling which has a stride of two. For up-sampling we use nearest neighbour. The number under each box shows the number of filters while the top numbers indicate the input size. To model the dynamics, we use two layers of LSTMs (Hochreiter & Schmidhuber, 1997).

## 3 THE FITVID MODEL

Striving for simplicity, we propose the FitVid model for stochastic video prediction, a convolutional non-hierarchical variational model with a fixed prior. In this section we will describe the proposed model and arcitecture which is also visualized in Fig 2. The appendix includes more architecture details and the code is available in Supplementary Material.

### 3.1 VARIATIONAL VIDEO PREDICTION WITH FIXED PRIOR

Following prior work (Finn et al., 2016a; Babaeizadeh et al., 2018; Denton & Fergus, 2018; Rubinstein, 1997; Oprea et al., 2020; Wu et al., 2021a), we define the problem of pixel-level video prediction as follows: given the first $c$ frame of a video $\mathbf{x}_{<c} = \mathbf{x}_0, \mathbf{x}_1, \ldots, \mathbf{x}_{c-1}$, our goal is to predict the future frames by sampling from $p(\mathbf{x}_{c:T}|\mathbf{x}_{<c})$. Optionally, the predictive model can be conditioned on additional given information $\mathbf{a}_t$, such as the actions that the agents in the video are planning to take. This is typically called action-conditioned video prediction.

Variational video prediction (Babaeizadeh et al., 2018) follows the variational auto-encoder (Kingma & Welling, 2014) formalism by introducing a set of latent variables $\mathbf{z}$ to capture the inherent stochasticity of the problem. We factorize the likelihood model to $\prod_{t=c}^{T} p_\theta(\mathbf{x}_t|\mathbf{x}_{<t}, \mathbf{z}_{\leq t})$ which is parametrized in an autoregressive manner over time; i.e. at each timestep $t$ the video frame $\mathbf{x}_t$ and the latent variables $\mathbf{z}_t$ are conditioned on the past latent samples and frames.

For inference, we need to compute a marginalized distribution over the latent variables $\mathbf{z}$, which is intractable. To overcome this problem, we use variation inference (Jordan et al., 1999) by defining an amortized approximate posterior $q(\mathbf{z}|\mathbf{x}) = \prod_t q(\mathbf{z}_t|\mathbf{z}_{<t}, \mathbf{x}_{\leq t})$ that approximates the posterior distribution $p(\mathbf{z}|\mathbf{x})$. The approximated posterior is commonly modeled with an inference network $q_\phi(\mathbf{z}|\mathbf{x})$ that outputs the parameters of a conditionally Gaussian distribution $\mathcal{N}(\mu_\phi(\mathbf{x}), \sigma_\phi(\mathbf{x}))$. This network can be trained using the reparameterization trick (Kingma & Welling, 2014) by optimizing the variational lower bound (Kingma & Welling, 2014; Rezende et al., 2014): $\mathcal{L}(\mathbf{x}) = -\mathbb{E}_{q_\phi(\mathbf{z}|\mathbf{x})}\left[\log p_\theta(\mathbf{x}_{t:T}|\mathbf{x}_{<t}, \mathbf{z})\right] + \beta D_{KL}\left(q_\phi(\mathbf{z}|\mathbf{x})||p(\mathbf{z})\right)$ where $D_{KL}$ is the Kullback-Leibler divergence between the approximated posterior and the fixed prior $p(\mathbf{z}) = \mathcal{N}(\mathbf{0}, \mathbf{I})$. The hyper-parameter $\beta$ represents the trade-off between minimizing frame prediction error and fitting the prior (Higgins et al., 2016b;a; Denton & Fergus, 2018). For FitVid, we set $\beta = 1$.

### 3.2 THE PROPOSED ARCHITECTURE

**Encoder and decoder.** Following the recent advances in image generation, we use similar residual encoding and decoding cells as NVAE (Vahdat & Kautz, 2020). Each cell includes convolutional

layers with batch-normalization (Ioffe & Szegedy, 2015) and swish (Ramachandran et al., 2017; Elfwing et al., 2018; Hendrycks & Gimpel, 2016) as the activation function, followed by Squeeze and Excite (Hu et al., 2018). The encoder is made of four encoding blocks with two cells in each block. There is down-sampling after each encoder block using a strided convolution of size three in the spatial dimensions. The decoder also consists of four decoding blocks with two cells in each block, and a nearest neighbour up-sampling after each block. The number of filters in each encoding block is doubled while the number of filters in each decoding block is halved from the previous one. There is a residual skip connection between the encoder and the decoder after each cell which are fixed to the output from the last context frame. The statistics for batch-normalization is averaged across time.

**Dynamics model.** Similar to Denton & Fergus (2018), the encoded frame $\mathbf{h}_t$ is used to predict $\mathbf{h}_{t+1}$ using two layers of LSTMs (Hochreiter & Schmidhuber, 1997). Likewise, $q(\mathbf{z}_t|\mathbf{x}_{<t})$ is also modeled using a single layer of LSTMs with $\mathbf{h}_{t+1}$ as the input that outputs the parameters of a conditionally Gaussian distribution $\mathcal{N}(\mu_\phi(\mathbf{x}), \sigma_\phi(\mathbf{x}))$. During the training, $\mathbf{z}$ is sampled from $q(\mathbf{z}_t|\mathbf{x}_{<t})$ while at the inference time $\mathbf{z}$ is sampled from the fixed prior $\mathcal{N}(\mathbf{0}, \mathbf{I})$. The input to the model is always the ground-truth image (which is usually referred to as teacher-forcing (Goodfellow et al., 2016; Cenzato et al., 2019)). At inference time, the predicted image in the previous time-step is used as input to predict the next frame.

**Data augmentation.** We find that FitVid can fit even very large datasets very well. However, in many cases, even for datasets that prior models struggle to fit effectively, we find that FitVid actually begins to suffer from *overfitting* (read Section 4 and 5). To prevent the model from overfitting we use data augmentation. To the best of our knowledge, this is the first use of augmentation in video prediction, perhaps because prior state-of-the-art models tend to underfit already and therefore would not benefit from it. Given the rich literature in image augmentation, we augment the videos using RandAugment (Cubuk et al., 2020). We randomize the augmentation per video but keep the randomization constant for frames of a single video. RandAugment substantially improves the overfitting, however not entirely, as it can be seen in Figure 3. We improve the augmentation by selecting a random crop of the video before resizing it to the desired resolution at the training time, called RandCrop. The combination of RandCrop and RandAugment successfully prevents the overfitting, leading to models that both fit the training set and generalize well to held-out videos.

**What FitVid does *not* need.** Prior works on variational video prediction (Finn et al., 2016b; Babaeizadeh et al., 2018; Lee et al., 2018; Villegas et al., 2019; Wu et al., 2021a), generally require a range of additional design decisions for effective training. Common design parameters include using curriculum training, commonly by scheduled sampling (Bengio et al., 2015), to mitigate distributional shift between training and generation time (Oh et al., 2015; Finn et al., 2016a; Lee et al., 2018; Cenzato et al., 2019); heuristically tuning $\beta$ in $\mathcal{L}(\mathbf{x})$ to balance the prediction vs fitting the prior (Denton & Fergus, 2018) by annealing it over the course of training (Babaeizadeh et al., 2018; Lee et al., 2018) or learned priors (Denton & Fergus, 2018; Villegas et al., 2019; Castrejon et al., 2019; Wu et al., 2021a). Each of these design choices introduces hyperparameters, tuning burden, and additional work when applying a model to a new task. FitVid does not require any of these details: we simply train FitVid by optimizing $\mathcal{L}(\mathbf{x})$ using Adam (Kingma & Ba, 2014).

## 4 EXPERIMENTS

To evaluate FitVid, we test it on four different real-world datasets and compare its performance with prior state-of-the-art methods, with comparable parameter count, using four different metrics. Our main goal is to demonstrate that FitVid can in fact fit on these datasets well and illustrate how augmentation can prevent FitVid from overfitting, resulting in state-of-the-art prediction performance. Please visit the anonymized website **sites.google.com/view/fitvid** to see samples of generated videos. The code to reproduce these experiments is available in Supplementary Material.

### 4.1 EXPERIMENTATION SETUP

**Datasets:** To test FitVid, we use four datasets that cover a variety of real-life scenarios. We use the Human3.6M dataset (Ionescu et al., 2014), which consists of actors performing various actions in a room to study the structured motion prediction. We also use the KITTI dataset (Geiger et al., 2013b) to evaluate FitVid's ability to handle partial observability and dynamic backgrounds. For

Table 1: The empirical comparison between FitVid (with 302M parameters), GHVAE (Wu et al., 2021a) (with 599M parameters) and SVG (Villegas et al., 2019) (with 298M parameters). To prevent FitVid from overfitting, we use augmentation for Human3.6M and KITTI. The green color highlights where FitVid achieved state-of-the-art result while the red color highlights otherwise.

| RoboNet | FVD↓ | PSNR↑ | SSIM↑ | LPIPS↓ |
|---|---|---|---|---|
| GHVAE | 95.2 | 24.7 | 89.1 | 0.036 |
| SVG | 123.2 | 23.9 | 87.8 | 0.060 |
| FitVid (ours) | 62.5 | 28.2 | 89.3 | 0.024 |

| KITTI | FVD↓ | PSNR↑ | SSIM↑ | LPIPS↓ |
|---|---|---|---|---|
| GHVAE | 552.9 | 15.8 | 51.2 | 0.286 |
| SVG | 1217.3 | 15.0 | 41.9 | 0.327 |
| FitVid (ours) | 884.5 | 17.1 | 49.1 | 0.217 |

| Human3.6M | FVD↓ | PSNR↑ | SSIM↑ | LPIPS↓ |
|---|---|---|---|---|
| Skip Frame of 1 | | | | |
| GHVAE | 355.2 | 26.7 | 94.6 | 0.018 |
| SVG | - | - | - | 0.060 |
| FitVid (ours) | 154.7 | 36.2 | 97.9 | 0.012 |
| Skip Frame of 8 | | | | |
| SVG | 429.9 | 23.8 | 88.9 | - |
| FitVid (ours) | 385.9 | 27.1 | 95.1 | 0.026 |

both datasets, we followed the pre-processing and testing format proposed by Wu et al. (2021a) and Villegas et al. (2019), which predicts 25-frames conditioned the previous five at $64 \times 64$ resolution.

To evaluate FitVid in an action-conditioned setting, we use RoboNet dataset (Dasari et al., 2019). This large dataset includes more than 15 million video frames from 7 different robotic arms pushing objects in different bins. It contains a wide range of conditions, including different viewpoints, objects, tables, and lighting. Prior video prediction methods have a tendency to badly underfit on this dataset (Dasari et al., 2019). We follow the same setup as Wu et al. (2021a) and randomly select 256 videos for testing. Similar to Wu et al. (2021a), we train FitVid to predict next ten frames given two context frames as well as the ten future actions.

Finally, to compare FitVid to a wider range of prior work, we use the BAIR robot pushing dataset (Ebert et al., 2017), which is a widely-used benchmark in the video prediction literature. We follow the evaluation protocol of Rakhimov et al. (2020), which predicts the next 16 frames given only one context frame and no actions. Given the high stochasticity of robotic arm movement in BAIR, and the variety of unseen objects in the test set, it is a great benchmark for evaluating the model's ability to generate diverse outputs.

**Metrics:** We evaluate our method and prior models across four different metrics: Structural Similarity Index Measure (SSIM) (Wang et al., 2004), Peak Signal-to-noise Ratio (PSNR) (Huynh-Thu & Ghanbari, 2008), Learned Perceptual Image Patch Similarity (LPIPS) (Zhang et al., 2018) and Fréchet Video Distance (FVD) (Unterthiner et al., 2018). FVD measures the overall visual quality and temporal coherence without reference to the ground truth video. PSNR, SSIM, and LPIPS measure pixel-wise similarity to the ground-truth with LPIPS most accurately representing human perceptual similarity. Given the stochastic nature of video prediction benchmarks, we follow the standard stochastic video prediction evaluation protocol (Babaeizadeh et al., 2018; Villegas et al., 2019; Wu et al., 2021a): we sample 100 future trajectories per video and pick the best one as the final score for PSNR, SSIM and LPIPS. For FVD, we use all 100 with a batch size of 256.

## 4.2 RESULTS

**Comparisons:** First, we compare FitVid to GHVAE (Wu et al., 2021a) and SVG (Villegas et al., 2019). GHVAE is a hierarchical variational video prediction model trained in a greedy manner. SVG is a large-scale variational video prediction model with learned prior and minimal inductive bias. We chose these two baseline because they both investigated overfitting by scaling the model, and achieve state-of-the-art results. However, SVG reported no overfitting even for their biggest model ($(M = 3, K = 5)$) with 298M parameters (Villegas et al., 2019) while GHVAE (with 599M paramteres) reported "some overfitting" on smaller datasets (Wu et al., 2021a). At the same time, both of these models share a similar architecture to FitVid.

Table 1 contains the results of these experiments. As it can be seen in this table, FitVid outperforms both SVG and GHVAE across all metrics in Robonet and Human3.6M. In KITTI, FitVid also consistently outperforms SVG while either improves or closely matches the performance of GHVAE which has more than twice as parameters. For qualitative results, see Figures 4, 5 and 6.

**Comparison to non-variational methods:** To provide a comprehensive comparison with prior methods, including non-variational models, we test FitVid on BAIR dataset (Ebert et al., 2017). As can be seen in Table 2, FitVid outperforms most of the previous models in this setting while performing comparably to Video Transformer (Weissenborn et al., 2019) with 373M parameters.

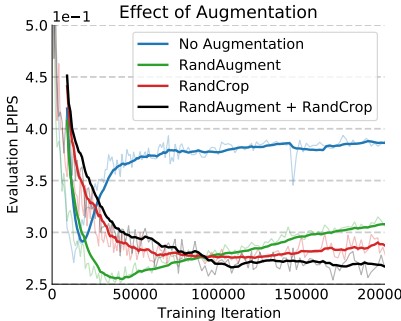

Figure 3: Effect of augmentation methods on overfitting on KITTI. Both RandAugment and Random Crop improve the overfitting but the combination of two gets the best results.

Table 2: Comparison between FitVid and different methods for video prediction on action-free BAIR dataset (Ebert et al., 2017).

| BAIR | FVD↓ |
|---|---|
| SV2P (Babaeizadeh et al., 2018) | 262.5 |
| LVT (Rakhimov et al., 2020) | 125.8 |
| SAVP (Lee et al., 2018) | 116.4 |
| DVD-GAN-FP (Clark et al., 2019) | 109.8 |
| VideoGPT (Yan et al., 2021) | 103.3 |
| TrIVD-GAN-FP (Luc et al., 2020) | 103.3 |
| cINNs (Dorkenwald et al., 2021) | 99.3 |
| CCVS (Moing et al., 2021) | 99.0 |
| Video Transformer (Weissenborn et al., 2019) | 94.0 |
| FitVid (ours) | 93.6 |

## 5 ANALYSIS

In this section, we take a closer look at the results from Section 4, to analyse the consequences of overfitting and the effect of regularization on the current benchmarks.

**On Human3.6M as a video prediction benchmark:** Human3.6M (Ionescu et al., 2014) is a common benchmark in video prediction literature (Finn et al., 2016a; Babaeizadeh et al., 2018; Villegas et al., 2017; Wang et al., 2019b; Villegas et al., 2019; Lin et al., 2020; Franceschi et al., 2020; Guen & Thome, 2020; Wu et al., 2021a;c) which we also use to evaluate FitVid (Figure 6). At the first glance, it seems that the model is generating extremely detailed and human-like motions conditioned on the given context pose. However, on closer inspection, we observe that the human subject in the predicted video is changing. In fact, FitVid replaces the unseen human subject into a training subject which is particularly evident from the clothing. Actually, we can find similar video clips from the training data for each one of the predicted videos (see Figure 6). These frames are **not** exactly the same, but they look notably similar. This observation indicates that:

1. The model can **generalize** to unseen frames, poses and subjects since the test context frames are new and unseen. FitVid correctly distinguished the subject from the background, detects the human pose and continues the video from there.
2. The model **memorized** the motion and the appearance of the training subjects. The model *morphs* the test human subject into a training one, and then *plays* a relevant video from the memory.

This means that FitVid fails to generalize to a new subject, while still generalizing to unseen frames. Given that the Human3.6M has five training and two test subjects (Finn et al., 2016a; Villegas et al., 2017) this may not be surprising.

Nevertheless, this observation shows how the current low-resolution setup for Human3.6M is not suitable for large-scale video prediction. In fact, after this observation, we traced the same behaviour in other video prediction literature and, unfortunately, it seems this is a common and overlooked issue. For example, the same phenomena can be seen in Figure 6 from Franceschi et al. (2020) that shows changing the test to a training subject by Struct-VRNN (Minderer et al., 2019) and the proposed method by Franceschi et al. (2020) (note the changed shirt). For others examples, see Figure 7 of Villegas et al. (2019) and Figure 5 of Villegas et al. (2017). See Figure 15 for a copy of these figures.

**Overfitting and regularization:** As mentioned in Section 1, there is considerable evidence that current video prediction models tend to underfit when trained on large datasets (Villegas et al., 2019; Castrejon et al., 2019; Wu et al., 2021a). Wu et al. (2021a), which is the current state-of-the-art model with 599 million parameters, reported "some overfitting" on smaller datasets such as Human3.6M and KITTI. However, we observe severe and clear overfitting with FitVid, despite having only 302 million parameters. Figure 7 visualizes the training and evaluation LPIPS metric while training FitVid on Human3.6M, **without augmentation**. This graph demonstrates that the training keeps getting better while the test quality starts to get worse after ∼15K iterations. We also observed similar behaviour on KITTI, as can be seen in Figure 7b. These results clearly shows that FitVid is overfitting on Human3.6M and KITTI, indicating that FitVid is using its parameters more efficiently. As mentioned

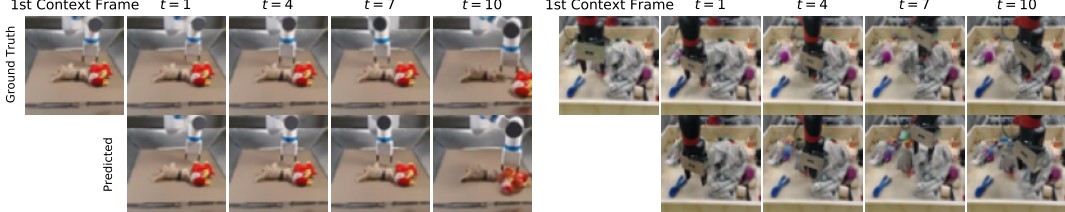

Figure 4: FitVid on action-conditioned RoboNet (Dasari et al., 2019). The model is conditioned on the first two frames and is predicting the next ten frames given the future actions of the robotic arm. These figures demonstrate how the predicted movements of the arm closely follows the ground truth given that the future actions is known. The model also predicts detailed movements of the pushed objects (visible in the left example) as well as filling in the previously unseen background with some random objects (look at the object that appear behind the robotic arm in the right). Also, to demonstrate a failure case, notice the wrong predictions of robot's fingers in the right example. See Figure 11 for more frames from these video samples.

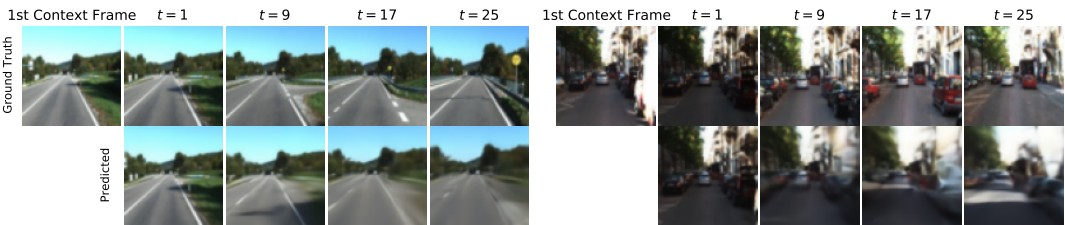

Figure 5: FitVid on KITTI dataset (Geiger et al., 2013b). As it can be seen in this figure, the model generates high quality prediction of the future in a dynamic scene. Note how in the top example FitVid keeps predicting the movement of the shadow on the ground till it gets out of the frame. After that, the model still brings the background closer in each frame, implying driving forward. We noticed that the quality of predictions drop substantially faster when there are more objects in the scene e.g. the driving scenes inside a city as can be seen in the right example. This indicates the model still fails to generalize to more complex scenes with more moving subjects. See Figure 12 for more frames.

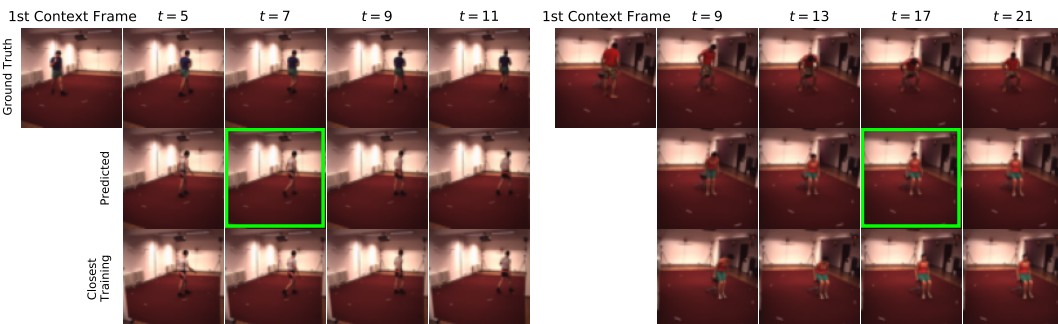

Figure 6: FitVid on Human3.6M (Ionescu et al., 2014). This figures demonstrates extremely detailed and human-like motions predicted by FitVid, conditioned on the given context frames. However, on closer inspection, it can be seen that the human subject in the video is changing, from the test subject to a training subject. This is particularly evident from the cloths. This phenomena indicates that, although FitVid is capable of generalizing to the frames out of training distribution, however, it morphs the human subject into a familiar one from the training set and then **plays** the video from the **memory**. In fact, we can find similar videos in the training set as visualized in the last row. The highlighted frame is the one used for finding the closest training video. Check Figure 13 for more predicted frames.

More videos can be found at **sites.google.com/view/fitvid**.

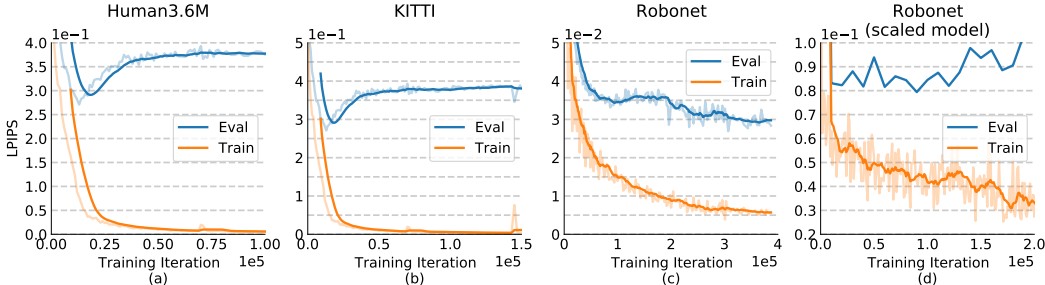

Figure 7: Overfitting of FitVid without augmentation. This figure visualizes the training and evaluation metrics on **(a)** Human3.6M (Ionescu et al., 2014), **(b)** KITTI (Geiger et al., 2013b) and **(c)** Robonet (Dasari et al., 2019), without augmentation. As it can be seen, in all cases, FitVid overfits on the training data except for Robonet. This is evident from the evaluation measurement going up while the training keeps decreasing. In case of Robonet, FitVid with 302M parameters did **not** overfit but a scaled version of the model with 600M parameters did, as can be seen in **(d)**. $y$-axis is LPIPS. $x$-axis is the training iteration. The plots are smoothed with an average rolling window of size ten. The shadows are the raw non-smoothed values.

in Section 3.2, to address overfitting, we use augmentation which results in state-of-the-art prediction quality, as reported in Table 1.

**Overfitting on Robonet:** We did not observe any overfitting on RoboNet, which is expected given the fact that RoboNet is much larger compared to the other benchmarks. Trying to find a model that can overfit on RoboNet, we test a scaled version of FitVid with 500M parameters – which is still smaller compared to GHVAE with 599M parameters and reported no overfitting on this dataset. This scaled version of FitVid overfits on RoboNet, as demonstrated in Figure 7d. Note that we did **not** use this scaled version in the reported numbers of Table 1, which is generated using the 302M version. Our goal here was to demonstrate that a scaled version of FitVid can also use its parameters more efficiently, compared to prior models, leading to overfitting on even bigger datasets such as RoboNet.

**Effect of Augmentation on SVG**: There is a discrepancy between the input data for training the models in Section 4. FitVid is trained with augmentation while the baselines are trained without any augmentation which raises a question: can the better performance of FitVid be explained only by the augmentation? In other words, do the previous methods benefit from augmentation too? To answer this question, we retrain SVG with and without augmentation. As demonstrated in Table 3, SVG performs worse if trained with augmented data, supporting the claim that it is underfitting to the raw data. As a result, this experiment provides more support for FitVid truly overfitting on these datasets and therefore benefiting from augmentation. Please note that we included our SVG results without augmentation too, as we could not perfectly reproduce the numbers reported by Villegas et al. (2019).

Table 3: SVG (Villegas et al., 2019) with and without augmentation. This table shows that SVG does not benefit from augmentation because it is underfitting to the original data (Villegas et al., 2017).

| **Human3.6M** | FVD↓ | PSNR↑ | SSIM↑ | LPIPS↓ |
|---|---|---|---|---|
| Without | 389.55 | 27.4 | 93.7 | 0.041 |
| With | 429.25 | 23.0 | 87.1 | 0.094 |

| **KITTI** | FVD↓ | PSNR↑ | SSIM↑ | LPIPS↓ |
|---|---|---|---|---|
| Without | 1612.62 | 14.8 | 38.7 | 0.330 |
| With | 2051.67 | 14.4 | 36.0 | 0.333 |

**Ablation Study** In order to figure out how much each design element in FitVid is contributing to its performamce, we ablate it and check its performance on RoboNet. In this study we try to cover a variety of ablations including encoder/decoder ablations, adding or removing modeling features such as learned prior as well as various skip connection implementations. As can be seen in Table 4, skip connections are a crucial component for archiving high prediction performance. This table also demonstrates that, while modeling sotchasticity is important even in action conditioned settings, FitVid does not benefit from a learned prior. For more ablation studies, please look at Section A.1.

Table 4: Ablation study of FitVid on RoboNet. This table demonstrates how much each design choice in FitVid contributes to its performance.

| | LPIPS↓ | PSNR↑ | SSIM↑ |
|---|---|---|---|
| Baseline (FitVid) | 0.024 | 28.2 | 0.89 |
| No Skip Connections | 0.088 | 22.3 | 0.75 |
| Non-residual Connections | 0.023 | 28.4 | 0.90 |
| Deterministic | 0.025 | 28.3 | 0.90 |
| With Learned Prior | 0.024 | 28.2 | 0.90 |
| ReLU activation | 0.062 | 23.2 | 0.81 |
| No Batch Norm | 0.771 | 6.0 | 0.00 |
| No Squeeze and Excite | 0.124 | 19.4 | 0.68 |

**Zero-shot Real Robot Performance:** Prior work indicate that improved video prediction translates to better performance in the downstream tasks (Wu et al., 2021a; Babaeizadeh et al., 2020). However, in these works, the training and test distribution are the same and there is almost no domain shift from training to testing. In this section, we are interested in investigating whether FitVid is capable of generalizing to a similar but visually different task with no training data for this new domain. Therefore, we setup a real-robot experiment, with a Franka Emika Panda robot arm, in which the goal is to push a specific object to a predetermined goal position. We train FitVid on RoboNet and use cross-entropy method (CEM) (Rubinstein, 1997; Chua et al., 2018) for planning (please see Appendix for details). As can be seen in Table 5, this agent is unable to generalize to the new domain, achieving worse performance than a random agent. This may not be surprising given the fact that the videos in RoboNet have entirely different robots and visuals, although the robots are performing the same task (i.e. pushing objects in a bin using a robotic arm).

Table 5: Zero-shot real robot performance. We use FitVid for planning future actions of a real robot pushing an object to a goal location with no training data from our setup. We train the model on visually different data (RoboNet) and the data from a closer domain (from Wu et al. (2021a)) with and without augmentation. While unable to directly adapt from RoboNet to the new domain, the results illustrate that fine-tuning on similar data and augmentation improve FitVid's performance.

| Training Data | Success Rate |
|---|---|
| Baseline (random actions) | 28% |
| RoboNet | 17% |
| RoboNet + Wu et al. (2021a) | 56% |
| RoboNet + Augmented Wu et al. (2021a) | 78% |

We then try to bring the training and test domain closer to each other by fine-tuning FitVid on the data from Wu et al. (2021a). This data contains 5000 autonomously collected videos of a Franka Emika Panda robot arm pushing objects around which look more similar to our setup compared to RoboNet, but still contain different lighting, camera angle, and target objects. This time, we observe that FitVid is relatively successful at generalizing to the new domain, succeeding in 56% of the trials. Finally, we find that adding data augmentation to the fine-tuning improves the generalization ability of the model, achieving 78% success rate. These results illustrate that while large distribution shift adaptation (RoboNet) remains difficult, by using data augmentation FitVid is capable of adapting to a relatively new domain (from Wu et al. (2021a) data).

## 6 CONCLUSION

We propose FitVid, a simple and scalable variational video prediction model that can attain a significantly better fit to current video prediction datasets even with a similar parameter count as prior models. In fact, while prior methods generally suffer from *underfitting* on these datasets, naïvely applying FitVid actually results in overfitting. We therefore propose using a set of existing image augmentation techniques that prevent overfitting in video prediction settings, leading to state-of-the-art results across a range of prediction benchmarks.

To the best of our knowledge, this is the first time a model reports, not only not underfitting, but rather substantial overfitting on these benchmarks. This is particularly important because underfitting is usually cited as one the main reasons for low quality predictions of the future frames. We demonstrate that simple existing image augmentation techniques can be used to prevent the mode from overfitting, resulting in high quality images. In fact, while each of the components of our proposed model are not novel on their own, FitVid outperforms the current state-of-the-art models across four different video prediction benchmarks on four different metrics. Moreover, the fit attained by FitVid is so good that a number of previously used benchmarks and metrics can be fooled, resulting in undesired outcomes, which are often overlooked in the video prediction literature.

There are many ways that FitVid can be expanded. As mentioned in the text, one of the interesting features of our proposed method is that it is simple. It is non-hierarchical, convolutional, with no attention mechanism, no curriculum learning, and no training scheduling. Any of these features can potentially improve the results of FitVid in order to generate even higher quality images. Given the simplicity of FitVid, it can be easily built upon. Another interesting direction would be to introduce new *training-aware* metrics for video prediction and generation to signal when a model is generating high quality videos by repeating the training data.

## ETHICS STATEMENT

Videos are an abundant source of visual information about our physical world. They contain information about objects, humans and how they interact with each other. The goal of video prediction is to foremost learn a representation of the world, usable for downstream tasks by an agent. Second, is to predict what happens next, conditioned on the past and the future intents, which can be used for planning. Despite many recent advances in this field, the present day models are still relatively low-quality and limited to narrow domains which makes their applications limited. However, if improved, video prediction or representations learned by video prediction, can be a major step forward toward fully autonomous self-learning agents. This paper, we believe, takes an important step towards this goal by pushing state-of-the-art forward and simplifying it in a meaningful way. However, our model is vulnerable to the bias in the training data and if adopted widely, this can skew research in certain directions. For example, our results may lead to higher quality models which can be scaled to generate even higher quality results. These models will be harder to design and train and require more computational power, and potentially can be biased. Finally, the underlying techniques for video prediction can be misused for generating high-quality videos that are misleading, depicting deliberately false situations or persons, similar to the phenomenon of deepfakes (Korshunov & Marcel, 2018). However, we believe that our experiments are conducted in relatively specific narrow settings and conditions which will likely not generalize broadly. Specifically, it is especially challenging to generate high-quality realistic-looking videos even with such state of the art methods.

## REPRODUCIBILITY STATEMENT

The anonymized implementation of the model, as well as the code to train and evaluate FitVid is attached in the Supplementary Material. This code can be used to regenerate the results reported in this paper. The approximated computation cost of rerunning these experiments is available in Section A.2.1. Appendix also includes details of implementation such as the training and evaluation details in Section A.2 as well as augmentation details in Algorithm 3. The detailed architecture of the encoder networks can be found in Table 9, decoder in Table 11, and the dynamic model in Table 10 of the Appendix. The hyper-parameters used are reported in Table 8. Table 13 includes the license of the benchmark datasets.

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

## A  APPENDIX

### A.1  MORE ABLATION STUDIES

In order to demonstrate how the design decisions behind FitVid's architecture affect its performance, we conduct more ablation studies. In the first study, we replace FitVid's encoder and decoder blocks with the blocks from SVG, while keeping everything else the same, and report the prediction performance on RoboNet in Table 6. From this table, it can be seen that the new encoder and decoder blocks are much more efficient at modeling the videos, resulting in improved prediction metrics. Please note that the numbers reported for SVG are different from Table 1, because we used FitVid 's dynamic model in this study.

In another study, we analyze the effect of the number of encoding and decoding blocks on the performance of the model. We very the number of blocks from one to three and measure the prediction quality on RoboNet. As it can be seen in Table 7, it's not just about making the architecture bigger, but that we are actually benefiting from the particular design of our encoder and decoder i.e. having exactly two encoding blocks and two decoding blocks is the optimal spot for achieving the best results in a reasonable number of training iterations while maintaining the number of parameters in the same ballpark as previous work.

### A.2  IMPLEMENTATION DETAILS

In this section we describe the details of FitVid's architecture as well as training. Algorithm 1 and Algorithm 2 describe the high-level training and prediction process respectively. Look at Table 8 for the used hyper-parameters We used the same set of hyper-parameters across all experiments. As mentioned in Section 3.2, all of the hyper-parameters are fixed during the training and there is no scheduling. Table 9 and Table 11 include the detailed architecture of the encoder and the decoder. Table 10 describes the structure of dynamic and posterior networks. Finally, Table 12 describes the augmentations details.

#### A.2.1  COMPUTATION RESOURCES

We implement FitVid using Flax (Heek et al., 2020) library for JAX (Bradbury et al., 2018). We train FitVid on $4{\times}4$ TPUs (32 co-processors). Each training step (with global batch size of 128 or local batch size of 4) takes $\sim 1020$ milliseconds (i.e. 0.98 step per second). In parallel, we evaluate the model every 1000 training steps on a single V100 GPU which takes about $\sim 850$ milliseconds for a batch size of 128. The models are trained for one million training iterations which takes $\sim 12$ days to complete.

---

**Algorithm 1:** FitVid training.

**Input:** Number of context frames $c$
**Data:** Training frames $\mathbf{x}_{0:T}$ and actions $\mathbf{a}_{0:T}$
   // Encode all frames
1  **for** $t \leftarrow 0$ **to** $T$ **do**
2    |   $\mathbf{h}_t, \mathbf{C}_t \leftarrow,$ Encoder($\mathbf{x}_t$)
3  **end**
   // Prediction
4  $\mathbf{s}_p, \mathbf{s}_d \leftarrow \mathbf{0}, \mathbf{0}$ // Initialize states
5  **for** $t \leftarrow 0$ **to** $T$ **do**
    | // Approximate posterior
6    | $(\mu_t, \sigma_t), \mathbf{s}_p \leftarrow$ Posterior($[\mathbf{h}_{t+1}], \mathbf{s}_p$)
7    | $\mathbf{z}_t \sim \mathcal{N}(\mu_t, \sigma_t)$
    | // Predict future state
8    | $\hat{\mathbf{h}}_t, \mathbf{s}_d \leftarrow$ Dynamic($[\mathbf{h}_t, \mathbf{a}_t, \mathbf{z}_t], \mathbf{s}_d$)
9  **end**
   // Decode all frames
10 **for** $t \leftarrow 0$ **to** $T$ **do**
    | // Use last available skip
    |     connection
11   | $\hat{\mathbf{x}}_t \leftarrow$ Decoder($\hat{\mathbf{h}}_t, \mathbf{C_c}$)
12 **end**
   // Optimize ELBO
13 $\mathcal{L} \leftarrow ||\mathbf{x} - \hat{\mathbf{x}}||_2 + D_{KL}\big(\mathcal{N}(\mu, \sigma), \mathcal{N}(\mathbf{0}, \mathbf{I})\big)$
14 $w \leftarrow$ Adam$\big(w, \mathcal{L}\big)$

---

**Algorithm 2:** FitVid prediction.

**Input:** Context frames $\mathbf{x}_{0:c}$
**Input:** All actions $\mathbf{a}_{0:T}$
**Output:** $\hat{\mathbf{x}}$
1  $\mathbf{s}_d \leftarrow \mathbf{0}$ // Initialize states
2  $, \mathbf{C} \leftarrow$ Encoder($\mathbf{x}_c$) // Get last skips.
3  **for** $t \leftarrow 0$ **to** $T$ **do**
    | // Encode frame
4    | $\mathbf{h}_t, \mathbf{C}_t \leftarrow,$ Encoder($\mathbf{x}_t$)
    | // Sample from prior
5    | $\mathbf{z}_t \sim \mathcal{N}(\mathcal{N}(\mathbf{0}, \mathbf{I}))$
    | // Predict future state
6    | $\hat{\mathbf{h}}_t, \mathbf{s}_d \leftarrow$ Dynamic($[\mathbf{h}_t, \mathbf{a}_t, \mathbf{z}_t], \mathbf{s}_d$)
    | // Decode frame
7    | $\hat{\mathbf{x}}_t \leftarrow$ Decoder($\hat{\mathbf{h}}_t, \mathbf{C_c}$)
8  **end**

---

Table 6: Effect of FitVid's encoder and decoder on prediction performance. As it can be seen in this paper, by replacing SVG's encoder and decoder with FitVid's encoder and decoder, the prediction quality improves.

| Encoder | Decoder | FVD↓ | LPIPS↓ |
|---------|---------|------|--------|
| SVG | SVG | 155.2 | 0.026 |
| FitVid | SVG | 113.1 | 0.025 |
| SVG | FitVid | 76.9 | 0.020 |
| FitVid | FitVid | **64.5** | **0.018** |

Table 7: Effect of the number of encoding and decoding blocks on the performance of FitVid. As it can be seen in this table, more encoder and decoder blocks do not essentially result in better prediction quality and having two blocks in both encoder and decoder is a good balance between number of paramters and the performance.

| Encoder Blocks | Decoder Blocks | FVD↓ | LPIPS↓ |
|---------------|----------------|------|--------|
| 1 | 1 | 121.3 | 0.022 |
| 1 | 2 | 100.8 | 0.021 |
| 1 | 3 | 73.3 | 0.019 |
| 2 | 1 | 100.3 | 0.025 |
| 2 | 2 | **64.5** | **0.018** |
| 2 | 3 | 176.6 | 0.028 |
| 3 | 1 | 88.8 | 0.019 |
| 3 | 2 | 83.9 | 0.019 |
| 3 | 3 | 78.6 | 0.018 |

Table 8: Hyper-parameters used for training FitVid. We used the same set of hyper-parameters across all experiments. As mentioned in Section 3.2, all of the hyper-paramters are fixed during the training and there is no scheduling.

| Hyper-parameter | Value |
|-----------------|-------|
| Optimizer (Adam(Kingma & Ba, 2014)) | |
| Learning Rate ($\alpha$) | 1e−3 |
| Btach Size | 128 |
| $\beta_1$ | 0.9 |
| $\beta_2$ | 0.999 |
| $\epsilon$ | 1e−8 |
| Gradient Clipping ($l_2$) | 100.0 |
| Model | |
| $\beta$ | 1 |
| Latent ($\mathbf{z}$)-dimension | 10 |
| Encoder ($\mathbf{h}$) dimension | 128 |
| LSTM size | 256 |

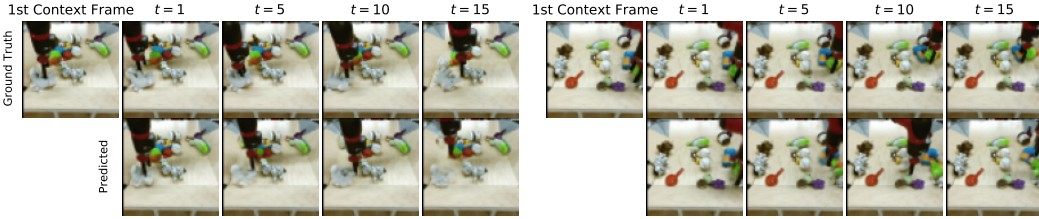

Figure 8: FitVid on BAIR robot pushing dataset (Ebert et al., 2017) with no actions. The model is conditioned only on the first frame and is predicting the next 16 frames. Given that the future actions of the robotic arm is unknown, the prediction can diverge substantially from the ground truth video. However, the model predicts movements for the objects whenever the arm pushes the object in an imaginary scenario. It also fills the background with random objects.

Table 9: FitVid Encoder Architecture. We are using the same encoding cells as NVAE (Vahdat & Kautz, 2020). The strides are always $1\times1$ except when down-sampling which has strides of $2\times2$. **(bn)** is batch-normalization (Ioffe & Szegedy, 2015), **(swish)** is the activation (Ramachandran et al., 2017), **(s&e)** is Squeeze and Excite (Hu et al., 2018). There is a skip connection from the beginning of each cell to the end of it. In these skip connections, the number of input filters will be matched by the output using a $1\times1$ convolution.

| Cell | Input Size | Pre | Kernel | Filters | Post | Down Sampling |
|------|-----------|-----|--------|---------|------|---------------|
| 1-1 | $64\times64$ | bn + swish | $3\times3$ | 64 | - | - |
| 1-1 | $64\times64$ | bn + swish | $3\times3$ | 64 | s&e | - |
| 1-2 | $64\times64$ | bn + swish | $3\times3$ | 64 | - | - |
| 1-2 | $64\times64$ | bn + swish | $3\times3$ | 64 | s&e | Yes |
| 2-1 | $32\times32$ | bn + swish | $3\times3$ | 128 | - | - |
| 2-1 | $32\times32$ | bn + swish | $3\times3$ | 128 | s&e | - |
| 2-2 | $32\times32$ | bn + swish | $3\times3$ | 128 | - | - |
| 2-2 | $32\times32$ | bn + swish | $3\times3$ | 128 | s&e | Yes |
| 3-1 | $16\times16$ | bn + swish | $3\times3$ | 256 | - | - |
| 3-1 | $16\times16$ | bn + swish | $3\times3$ | 256 | s&e | - |
| 3-2 | $16\times16$ | bn + swish | $3\times3$ | 256 | - | - |
| 3-2 | $16\times16$ | bn + swish | $3\times3$ | 256 | s&e | Yes |
| 4-1 | $8\times8$ | bn + swish | $3\times3$ | 512 | - | - |
| 4-1 | $8\times8$ | bn + swish | $3\times3$ | 512 | s&e | - |
| 4-2 | $8\times8$ | bn + swish | $3\times3$ | 512 | - | - |
| 4-2 | $8\times8$ | bn + swish | $3\times3$ | 512 | s&e | - |

Table 10: FitVid Dynamics Architecture. We are using a similar dynamics as Denton & Fergus (2018). The encoded output is first averaged across spatial dimension and then decoded into $\mathbf{h}$-size using a fully connected layer. Then, the dynamics are modeled by two LSTM layers. Finally, the output is mapped and reshaped to an image tensor before passing to the decoder. The posterior uses the exact same architecture except that only has one LSTM layer.

| Cell | Input Size | Pre | Layer | Size | Post |
|------|-----------|-----|-------|------|------|
| - | $8\times8$ | spatial average + flatten | dense | 256 | append $\mathbf{z}$ and $\mathbf{a}$ |
| - | 128+ | - | LSTM | 256 | - |
| - | 256 | - | LSTM | 256 | - |
| - | 256 | - | dense | $8\times8\times512$ | sigmoid + reshape |

Table 11: FitVid Encoder Architecture. We are using the same encoding and decoding cells as NVAE (Vahdat & Kautz, 2020). The strides are always $1{\times}1$. For up-sampling we use nearest neighbour. **(bn)** is batch-normalization (Ioffe & Szegedy, 2015), **(swish)** is the activation (Ramachandran et al., 2017), **(s&e)** is Squeeze and Excite (Hu et al., 2018). There is a skip connection from the beginning of each cell to the end of it. There are also skip connections from each encoder block to the corresponding decoder block (look at Figure 2). In these skip connections, the number of input filters will be matched by the output using a $1{\times}1$ convolution.

| Cell | Input Size | Pre | Kernel | Filters | Post | Up Sampling |
|------|-----------|-----|--------|---------|------|-------------|
| 1-1 | $8{\times}8$ | bn | $1{\times}1$ | 2048 | - | - |
| 1-1 | $8{\times}8$ | bn + swish | $5{\times}5$ | 2048 | - | - |
| 1-1 | $8{\times}8$ | bn + swish | $1{\times}1$ | 512 | bn + s&e | - |
| 1-2 | $8{\times}8$ | bn | $1{\times}1$ | 2048 | - | - |
| 1-2 | $8{\times}8$ | bn + swish | $5{\times}5$ | 2048 | - | - |
| 1-2 | $8{\times}8$ | bn + swish | $1{\times}1$ | 512 | bn + s&e | Yes |
| 2-1 | $16{\times}16$ | bn | $1{\times}1$ | 1024 | - | - |
| 2-1 | $16{\times}16$ | bn + swish | $5{\times}5$ | 1024 | - | - |
| 2-1 | $16{\times}16$ | bn + swish | $1{\times}1$ | 256 | bn + s&e | - |
| 2-2 | $16{\times}16$ | bn | $1{\times}1$ | 1024 | - | - |
| 2-2 | $16{\times}16$ | bn + swish | $5{\times}5$ | 1024 | - | - |
| 2-2 | $16{\times}16$ | bn + swish | $1{\times}1$ | 256 | bn + s&e | Yes |
| 3-1 | $32{\times}32$ | bn | $1{\times}1$ | 512 | - | - |
| 3-1 | $32{\times}32$ | bn + swish | $5{\times}5$ | 512 | - | - |
| 3-1 | $32{\times}32$ | bn + swish | $1{\times}1$ | 128 | bn + s&e | - |
| 3-2 | $32{\times}32$ | bn | $1{\times}1$ | 512 | - | - |
| 3-2 | $32{\times}32$ | bn + swish | $5{\times}5$ | 512 | - | - |
| 3-2 | $32{\times}32$ | bn + swish | $1{\times}1$ | 128 | bn + s&e | Yes |
| 4-1 | $64{\times}64$ | bn | $1{\times}1$ | 256 | - | - |
| 4-1 | $64{\times}64$ | bn + swish | $5{\times}5$ | 256 | - | - |
| 4-1 | $64{\times}64$ | bn + swish | $1{\times}1$ | 64 | bn + s&e | - |
| 4-2 | $64{\times}64$ | bn | $1{\times}1$ | 256 | - | - |
| 4-2 | $64{\times}64$ | bn + swish | $5{\times}5$ | 256 | - | - |
| 4-2 | $64{\times}64$ | bn + swish | $1{\times}1$ | 64 | bn + s&e | - |
| - | $64{\times}64$ | - | $1{\times}1$ | 3 | sigmoid | - |

Table 12: To prevent FitVid from overfitting we use augmentation. First, at training time, we select a random crop of the video before resizing it to the desired resolution ($64{\times}64$) at the training time, called RandCrop. This processes crops all the frames of a given video to include a minimum of $C$ percent of the frame's height. Then we use RandAugment (Cubuk et al., 2020) to improve the augmentation. We use the same augmentation configuration for all the datasets. Per video, we use the same randomization across all the frames.

---

**Algorithm 3:** Video Augmentation.

**Input:** Video $\mathbf{x}$
**Input:** Number of RandAugment transformations $N$
**Input:** RandAugment magnitude $M$
**Input:** RandCrop crop height minimum ratio $C$
1   $\mathbf{x} \leftarrow RandCrop(\mathbf{x}, C)$
2   **for** $i \leftarrow 0$ **to** $N$ **do**
3     $f \leftarrow$ ChooseRandomTransformation()
4     $\mathbf{x} \leftarrow f(\mathbf{x}, M)$
5   **end**
6   **return** $\mathbf{x}$

| Hyper-parameter | Value |
|-----------------|-------|
| **RandAugment** | |
| No. of transformations $N$ | 1 |
| Magnitude $M$ | 5 |
| **Transformations** | |
| identity - auto_contrast - equalize rotate - solarize - color - posterize contrast - brightness - sharpness shear_x - shear_y translate_x - translate_y | |
| **RandCrop** | |
| Crop height minimum ratio $C$ | 0.8 |

Table 13: Used datasets and their licenses.

| Dataset | Reference | License |
|---------|-----------|---------|
| RoboNet | Dasari et al. (2019) | MIT |
| KITTI | Geiger et al. (2013a) | Creative Commons |
| Human3.6M | Ionescu et al. (2014) | License |
| BAIR robot pushing | Ebert et al. (2017) | GitHub Default |

## Real Robot Domain

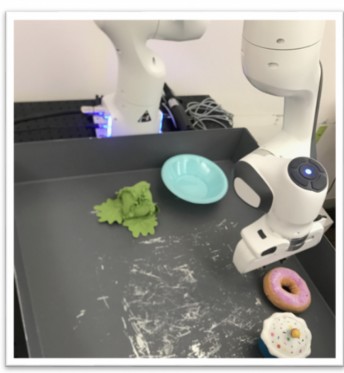 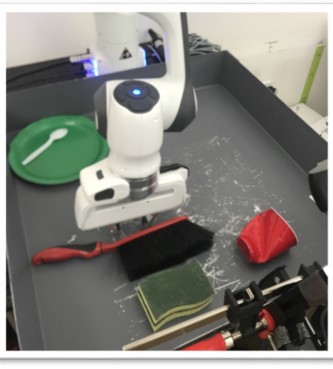 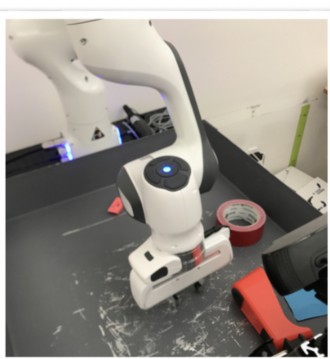

**Kitchen Items**          **Cleaning Tools**          **Office Supplies**

Figure 9: Zero-shot robot domain. We evaluate FitVid on planning tasks on a Franka Emika Panda involving kitchen, cleaning, and office items. We did not collect any training data for this task. Instead the model is trained on RoboNet and fine-tuned on augmented data from Wu et al. (2021a).

### A.3   ROBOT EXPERIMENT DETAILS

#### A.3.1   ENVIRONMENT

The robot environment consists of a Franka Emika Panda robot operating over a bin which contains various objects. The robot's observations are $64\times64\times3$ RGB images, and its action space consists of 3 DOF delta position control of the end effector with action magnitudes in the range [-10cm,10cm].

#### A.3.2   DATA

The data used for finetuning the model used in robot experiments was taken directly from Wu et al. (2021a). This data consists of different viewpoint, lighting conditions, and target objects than what is used in our evaluation.

#### A.3.3   EVALUATION DETAILS

During evaluation, the agent is specified to complete an object pushing task by a goal image. The agent has 50 timesteps to complete the task, and a trial is measured as successful if the majority of the object overlaps with its target position at some point in the episode. Each method is evaluated over 18 trials, of which 6 consist of objects in an "office" setting, 6 consist of objects in a "kitchen" setting, and 6 consist of objects in a "cleaning" setting (See Figure 9).

To execute the task, the agent performs visual model predictive control using the cross entropy method (CEM). Specifically, the agent takes in 1 frame, and predicts trajectories of 10 time-steps for 200 different sampled action sequences. Trajectories are ranked according to their negative mean squared error to the goal image, averaged across all 10 time-steps. The action distribution refits to the top 20 actions, and repeats for 3 iterations of CEM. Afterwards the best sequence of 10 actions is stepped in the environment in an open loop faction. The process repeats 5 times until the end of the episode.

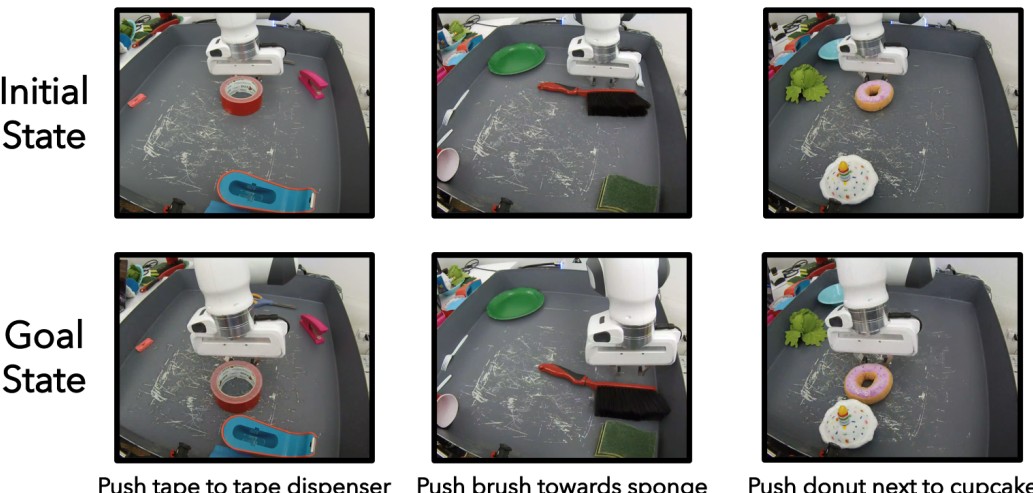

Initial State | Goal State

Push tape to tape dispenser     Push brush towards sponge     Push donut next to cupcake

Figure 10: Example tasks for zero-shot object pushing using a robotic arm. The goal in each trial is to push the a specific object to a predetermined goal location. The trial is considered successful, if the robot pushes at least half of the object overlaps with its goal location at any point in the episode.

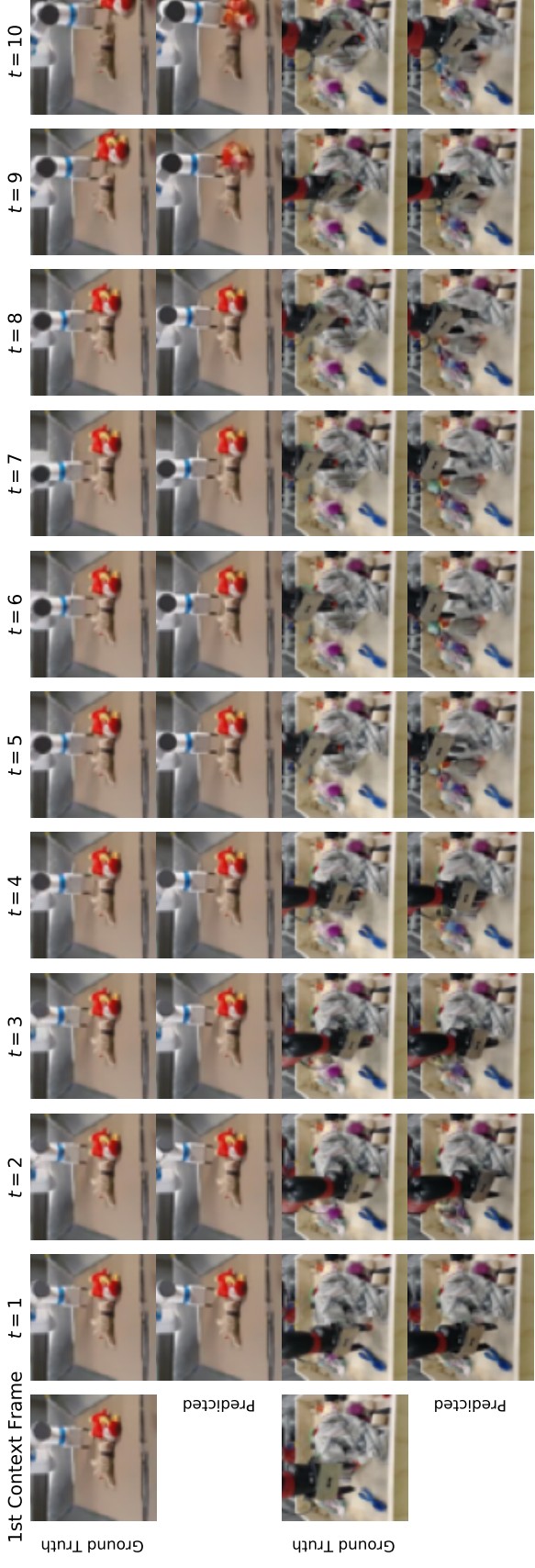

Figure 11: More detailed video from Figure 4 which illustrates FitVid on action-conditioned RoboNet (Dasari et al., 2019). The model is conditioned on the first two frames and is predicting the next ten frames given the future actions of the robotic arm. These figures demonstrate how the predicted movements of the arm closely follows the ground truth given that the future actions is known. The model also predicts detailed movements of the pushed objects (visible in the top example) as well as filling in the previously unseen background with some random objects (look at the object that appear behind the robotic arm in the bottom example). Also notice the wrong predictions of robots fingers in the bottom example.

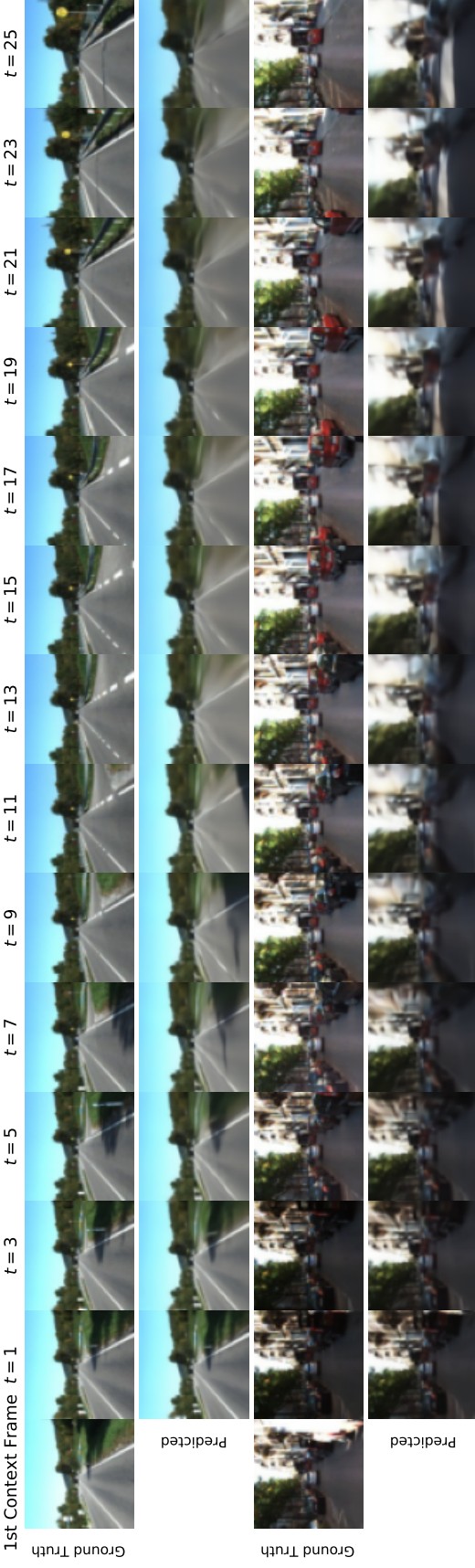

Figure 12: More detailed video from Figure 5 which illustrates FitVid on KITTI dataset (Geiger et al., 2013b). As it can be seen in this figure, the model generates high quality prediction of the future in a dynamic scene. Note how in the top example FitVid keeps predicting the movement of the shadow on the ground till it moves out of the frame. After that, the model keeps pushing the background closer in each frame, implying driving forward. We noticed that the quality of predictions drop substantially faster when there are more objects in the scene e.g. the driving scenes inside a city as can be seen in the bottom example. This indicates the model still fails to generalize to more complex scenes with more moving subjects.

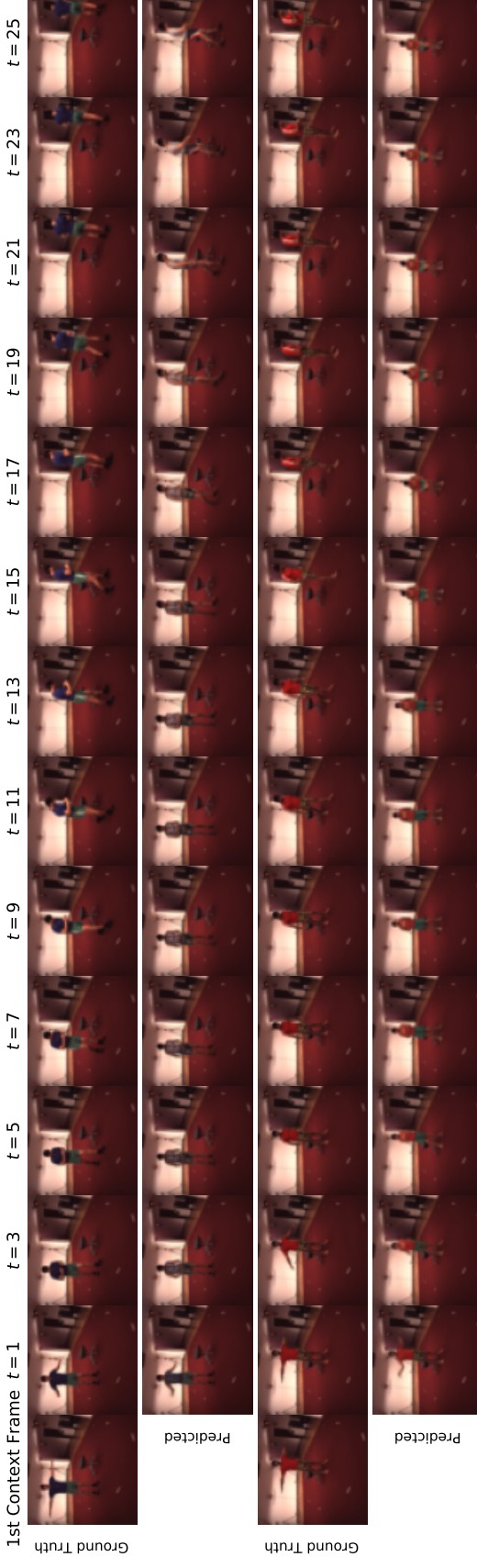

Figure 13: More detailed video from Figure 6 which illustrates FitVid on Human3.6M (Ionescu et al., 2014). This figure demonstrates extremely detailed and human-like motions predicted by FitVid, conditioned on the given context frames. However, on closer inspection, it can be seen that the human subject in the video is changing, from the test subject to a training subject. This is particularly evident from the cloths. This phenomena indicates that, although FitVid is capable of generalizing to the frames out of training distribution, however, it morphs the human subject into a familiar one from the training set and then plays the video from the memory.

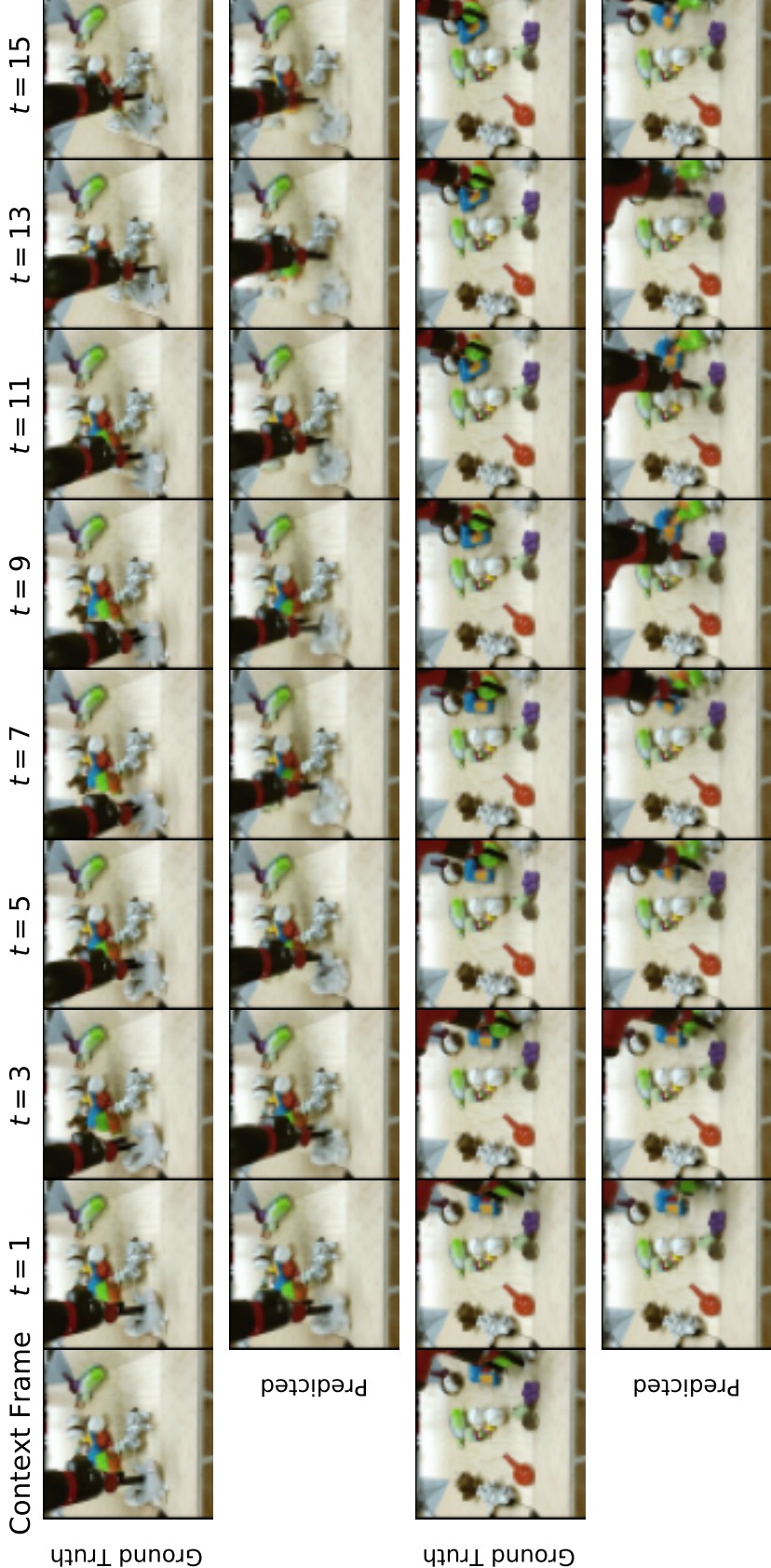

Figure 14: More detailed video from Figure 8 which illustrates FitVid on BAIR robot pushing dataset (Ebert et al., 2017) with no actions. The model is conditioned only on the first frame and is predicting the next 16 frames. Given that the future actions of the robotic arm is unknown, the prediction can diverge substantially from the ground truth video. However, the model predicts movements for the objects whenever the arm pushes the object in an imaginary scenario. It also fills the background with random objects.

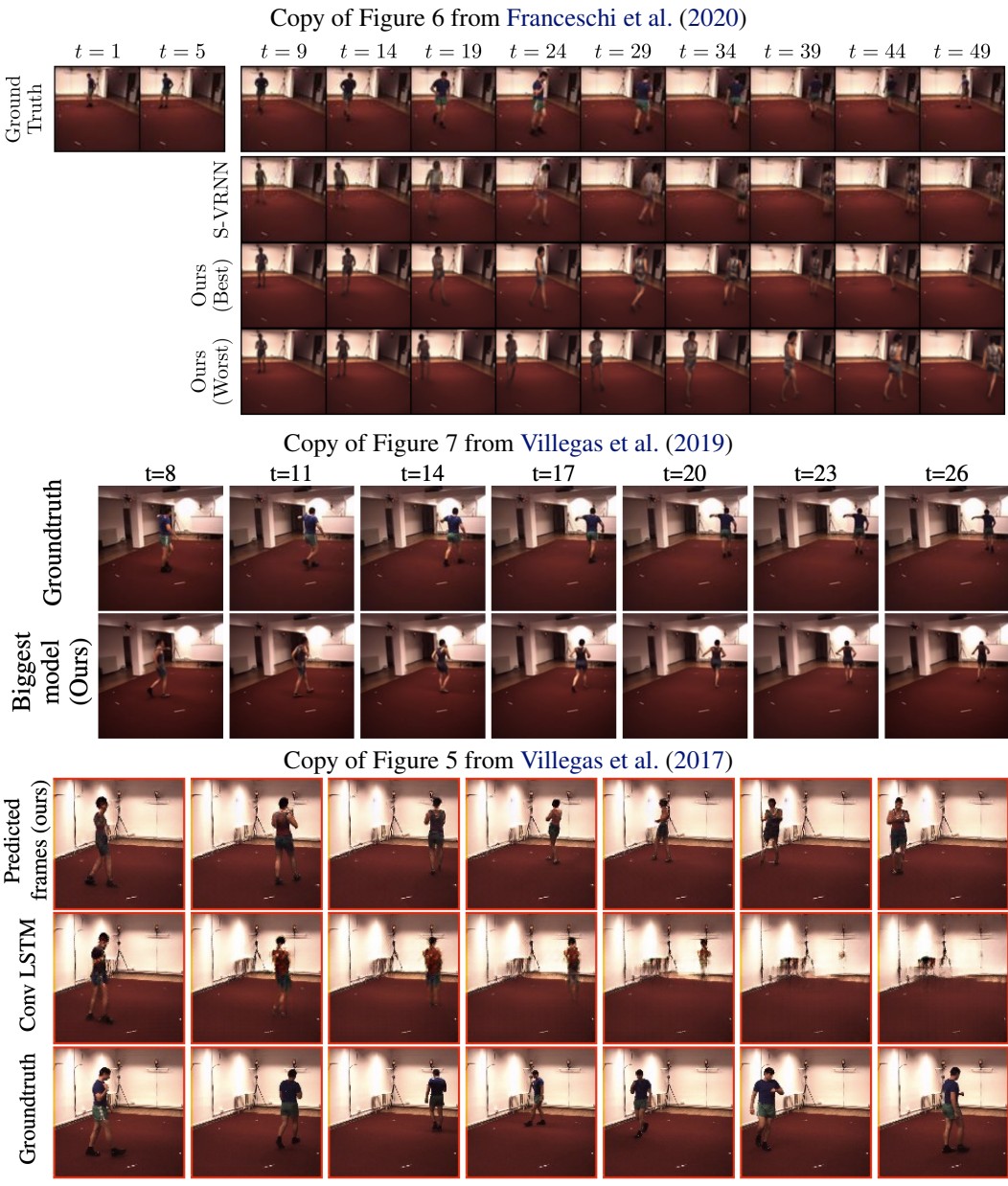

Figure 15: These are copies of Figure 6 from Franceschi et al. (2020), Figure 7 from Villegas et al. (2019) and Figure 5 from Villegas et al. (2017). The proposed methods in these papers are also changing the human test subject into a training subject (moslt visible in the changed shirt.). This seems to be a common issue which is typically overlooked in the video prediction literature.

