# OpenReview forum: "FitVid: High-Capacity Pixel-Level Video Prediction"
_ICLR.cc/2022/Conference — ICLR 2022 Submitted_

### Official Review · Reviewer_3nyU · 2021-11-01

**Correctness:** 3
**Technical Novelty And Significance:** 3
**Empirical Novelty And Significance:** 3
**Recommendation:** 6
**Confidence:** 2

**Main Review:**

The paper is easy to follow and is clearly written. The strengths can be summarized as:
+ The results look strong and are much better than state-of-the-art across a wide range of datasets and metrics. The qualitative results look particularly good.
+ The FitVid model is much easier to train and requires no tricks or extra designs to train as in previous works. It can be trained simply from scratch using Adam.
+ A main argument about the FitVid is that it does not suffer from underfitting issues and is even able to overfit. The ablation studies are extensive and indeed support this argument.

Weaknesses
- It is not clear how the FitVid model is different from previous models, and why it is able to overcome the underfitting issues while the previous work could not. Even without the data augmentation, FitVid seems to be able to overfit already. Why? More discussion should be devoted to this topic.
- Minor grammar issues. For example: "To provide more performance comparison more prior methods..."

It would also be great if the authors could provide some thoughts on this issue:
* Experiments on RoboNet show strikingly good results in Fig4. As stated in the figure caption -- "The model also predicts detailed movements of the pushed objects (visible in the left example) as well as filling in the previously unseen background with some random objects (look at the object that appear behind the robotic arm in the right)".   It seems miraculous to me that the model is able to figure out the random objects in the background and their appearance after a few frames. Usually such tasks require understanding of the physical scene (occlusion, object geometry), physics and even 3D rendering. It seems that the proposed model is able to figure out all these very high-level understanding of the scene using a relatively straightforward network design and no extra mechanisms. Could the authors shed some light on how this is achieved?

**Summary Of The Paper:**

This paper presents a new network architecture called FitVid to perform the task of video prediction, i.e. the task of predicting future frames from previous frames. Previous methods for this tasks usually suffer from "underfitting", while FitVid is able to overfit on major benchmarks without increasing the number of parameters, thanks to a better model architecture. What is more, FitVid is much easier to train than previous work, without using any bells and whistles in training. As a result, FitVid achieves state-of-the-art on four challenging video datasets across a wide range of metrics.

**Summary Of The Review:**

I think the paper could provide more insights on why the FitVid network is able to overfit without bells and whistles using even fewer params than previous methods. It would make the paper even stronger.


-------------------------------------------------------------------- Post rebuttal
I have carefully read the reviews and the rebuttal. The main concerns revolve around a lack of novelty. The authors rebut by providing a detailed study of the proposed components. I think the rebuttal addresses well why carefully designed training schedules are no longer necessary, which might be useful to the community. Novelty-wise, it is still a bit limited, and the overall impression to me it is a bit empirical. But given the strong performance, I am keeping my original rating.

---

> ### Author Response · Authors · 2021-11-16
> **Authors response.**
>
> We would like to thank the reviewer for their time and detailed review. Regarding effectiveness for each part of the FitVid and a more detailed comparison with prior work, please read our aggregated response in https://openreview.net/forum?id=iim-R8xu0TG&noteId=WEQXp8Kw5Ta
>
> > Minor grammar issues. For example: "To provide more performance comparison more prior methods..."
>
> Thank you for pointing it out. We fixed the issue.
>
> > Experiments on RoboNet show strikingly good results in Fig4. As stated in the figure caption -- "The model also predicts detailed movements of the pushed objects (visible in the left example) as well as filling in the previously unseen background with some random objects (look at the object that appear behind the robotic arm in the right)". It seems miraculous to me that the model is able to figure out the random objects in the background and their appearance after a few frames. Usually such tasks require understanding of the physical scene (occlusion, object geometry), physics and even 3D rendering. It seems that the proposed model is able to figure out all these very high-level understanding of the scene using a relatively straightforward network design and no extra mechanisms. Could the authors shed some light on how this is achieved?
>
> Thank you for such a detailed review, as well as the compliment about the quality of the results! :) It is important to notice that the predicted background does not match the reality. In fact, the model guesses that, given the high number of objects in the bin, there should be more behind the robotic arm. The model can make such a guess given “similar” training examples. I.e. in training videos, the model observed unseen objects appear behind the moving arm and repeats this behavior when generating new videos.
>
> That being said, the model has some clear notions of physics and how objects interact with each other. This is most evident in Figure 1, in which the model predicts the direction and the speed of the red plush object, when pushed by the robotic arm rather accurately. However, it stops moving this object after 5 frames guessing that the arm will slip from the bottom which does not happen in reality. At the same time, it predicts no movement for the khaki object which is correct. Such observations suggest that the model has a basic understanding of physics. In fact, there are some papers which explicitly explored this phenomenon e.g. https://arxiv.org/abs/1812.10972.
>
> So although we agree that the quality of generated videos on RoboNet is notably high, as it is one of the strongest attributes of FitVid, we don’t think it’s “miraculous”  :) It's rather generalizing from previously seen data points in the training data to unseen context frames.

---

### Official Review · Reviewer_7Y5j · 2021-11-01

**Correctness:** 4
**Technical Novelty And Significance:** 2
**Empirical Novelty And Significance:** 3
**Recommendation:** 5
**Confidence:** 4

**Main Review:**

Strengths:
The paper discovers the underfitting phenomenon which is a novel but the basic problem for the video prediction task. The FitVid's architecture is simple but effective. The author uses several existing methods to overcome the underfitting problem and achieves state-of-the-art performance on four challenging datasets. I really appreciate that the experiments of this paper are very solid.

Weakness:
The FitVid's architecture has no additional novelty and just combines the existing methods, such as SE, UNet, and LSTM. The paper did not analyze the effectiveness for each part of the FitVid. Why do previous methods not behave well? The explicit explanation of this problem is important and needs to be explored. I'm looking forward to your reply to this question.


====After rebuttal====
I have read the revised paper and rebuttal comments. I do really appreciate the author's effort to clarify the novelty of the combination of each module. However, I still hope that the proposed combination could be intuitively inspired by the overfit phenomena rather than you just design it and use a large number of experiments to vary its effectiveness.

**Summary Of The Paper:**

The paper proposes a simple and scalable variational video prediction model FitVid, which attains a better fit to video prediction datasets even with a similar parameter count as prior models. The author has observed that previous methods suffer from underfitting on these datasets, directly applying FitVid actually results in overfitting. The FitVid uses a set of existing image augmentation techniques to prevent overfitting so that it can achieve state-of-the-art results on several prediction benchmarks. FitVid's architecture is based on SE-UNet LSTM, which seems to be the common backbone for the stochastic video prediction task.

**Summary Of The Review:**

The paper proposes a simple and scalable variational video prediction model FitVid, which could overcome the underfitting problem better.

---

> ### Author Response · Authors · 2021-11-16
> **Authors response.**
>
> We would like to thank the reviewer for their time and detailed review. Regarding effectiveness for each part of the FitVid and a more detailed comparison with prior work, please read our aggregated response in https://openreview.net/forum?id=iim-R8xu0TG&noteId=WEQXp8Kw5Ta

---

> ### Author Response · Authors · 2021-11-19
> **Recommendation**
>
> We noticed that you changed your recommendation of the paper without engaging in the discussion. Given the limited discussion period, we really appreciate it if we can hear your concerns and how our rebuttal response could be expanded to properly address them.
>
> To reiterate, we tried to address your main concerns, regarding the novelty and short comings of the previous methods, in the aggregated response here: https://openreview.net/forum?id=iim-R8xu0TG&noteId=WEQXp8Kw5Ta
>
> We hope that these additional ablations and comparisons help readers understand the importance of each of the individual design decisions in our architecture. If you believe that these experiments still do not sufficiently address the question of which components of FitVid are important or how/why FitVid works, we would really appreciate some recommendations about which additional evaluations would better answer this question. We are happy to run additional experiments and evaluations to get to the bottom of this question, but at the same time we think it's important to recognize that, when it comes to research on improving neural network architectures, there is only so much that can be done to fully understand the precise reasons for improvements, and we believe our efforts are well in line (and indeed in excess of) comparable work on better neural network architectures.
>
> Thank you.

---

### Official Review · Reviewer_exmV · 2021-11-02

**Correctness:** 3
**Technical Novelty And Significance:** 2
**Empirical Novelty And Significance:** 3
**Recommendation:** 5
**Confidence:** 4

**Main Review:**

Strength:
+ Scale is an important aspect of the system. The paper did a good job on investigating the overfit / underfit effect in their model.  Fig 6, analysis on Human 3.6M, Fig7, etc.
+ The experiments are conducted across 4 datasets under different domain which indicates the proposed method generalizes to some extent.
+ Though the generalization ability remains further exploration, a robot experiment based on MPC shows that their prediction model can be deployed in downstreaming tasks.

Weakness:
- Lack of Intuition. The author proposes an Unet like architecture that is claimed to use its parameters more efficiently. However, The intuition behind it is very limited to me. Why is theirs better? Where does the improvement come from? (they also mention they share similar architecture with baselines on Page5 Sec4.2 Para1)  Without intuition, the architecture design looks arbitrarily stacking blocks with some skip connections. I acknowledge that video predictions at pixel-level are extremely challenging and many have proposed their idea like only predicting transformation / hierarchical output / foreground-background factorization, etc.  But this work seems brute-force generates pixels with skip layers. I don’t see why it uses parameters more efficiently.  I look forward to authors justifying their idea.
- Limited Novelty. Though the paper has achieved some empirical contribution (manage to overfit and improve results on large-scale datasets), the technical contribution is limited. Their architecture is a UNet-based VAE. Data augmentation is well known to prevent overfitting and their particular augmentation is also based on prior work.
- Some heterogeneous metrics. Tabel1 does not report results on BAIR pushing dataset while Table2 only reports on BAIR pushing dataset on one metric. I wonder the reason behind the inconsistency. 1) can table 1 include results for BAIR? 2) can table 2 include other datasets? 3) can other metrics be included? E.g. report the best of 100 samples to incorporate with random sample.
- Their qualitative results on the website look very similar to baselines (GHVAE) for me.
- Claim on Sec 5 about generalizing to unseen frames. It is a weird claim about video prediction as the unseen future itself is output. The model copy-pasting something from train set does not indicate generalization.  I would just call it overfitting/memorizing train set. (okay w claim 2 but not claim 1)
- Is the figure 6 results with or without data augmentation? If it’s without, will data augmentation alleviate overfit? If it is with data augmentation, direct comparison (either quantitatively or qualitatively) is expected.


--------- after rebuttal -----
I'd like to thank authors for their detailed response and additional experiments/ablation that they ran. Some of my concerns are resolved (point 3,5,6). For point 4 (qualitative results), I cannot find qualitative comparison on the website anymore. I think it was there by the time I reviewed but not 100% sure. So I check the qualitative results in papers and on GHVAE's website. Most of figures in the paper only shows ground truth with their results w/o comparison except fig 15. For seqs on their and GHVAE website, I'm not convinced about the qualitative improvement, esp not the same seq. Quantitative results are important but qualitative comparison complements paper's arguments. For point 1,2, I appreciate the extra ablations by the authors. There are more empirical results (e.g. switching enc/dec respectively, hyperparam sweep) but the overall novelty is still limited. I'll keep my original score.

**Summary Of The Paper:**

The paper studies conditional video prediction. In particular, it focuses on the problem of current models underfitting and not scaling to datasets. The paper proposes an architecture that is claimed capable of using parameters more efficiently in order to overfit. Then, data augmentation is introduced to improve performance for generalization. They evaluated their method against baselines on 4 datasets (Human 3.6M, KITTI, Robonet, BAIR pushing dataset). They also showed experiments to support overfitting/underfitting claims.


**Summary Of The Review:**

The paper focuses on an important problem on scaling video prediction and manage to overfit to large-scale data as the first step followed by improving generalization by data augmentation. However, it is not clear to me why and how their proposed method is crucial to achieving this.

---

> ### Author Response · Authors · 2021-11-16
> **Authors response.**
>
> We would like to thank the reviewer for their time and detailed review. Regarding intuition please read our aggregated response in https://openreview.net/forum?id=iim-R8xu0TG&noteId=WEQXp8Kw5Ta
>
> > Limited Novelty. Their architecture is a UNet-based VAE. Data augmentation is well known to prevent overfitting and their particular augmentation is also based on prior work.
>
> We kindly disagree. Reducing the novelty of such paper to a “UNet-based VAE” is undermining the novelty of a new architecture, considering the fact that many impactful papers of the past can be categorized as such. The main baselines of this paper (GHVAE (CVPR 2021) and SVG (NeurIPS 2019)) are also “UNet-based VAEs” which are all published in top tier venues. However, all of these papers had a great impact on the field.
>
> Regarding data augmentation, please note that while data augmentation is standard in computer vision, it is **never** (to the best of our knowledge) used in video prediction papers and therefore its applications in video prediction is not obvious to begin with. The novelty of this paper is not to propose a new data augmentation for no good reason, but rather empirically demonstrating that existing image augmentation techniques are sufficient to prevent video prediction models from overfittting, which was unknown before this paper.
>
> > Some heterogeneous metrics. Tabel1 does not report results on BAIR pushing dataset while Table2 only reports on BAIR pushing dataset on one metric. I wonder the reason behind the inconsistency. 1) can table 1 include results for BAIR? 2) can table 2 include other datasets? 3) can other metrics be included? E.g. report the best of 100 samples to incorporate with random sample.
>
> For Robonet, Human3.6M and KITTI, the main baselines (GHVAE and SVG) provided detailed numbers across all 4 metrics. However, our goal in Table 2 was to provide more comparison with a wider list of previous models. Given the long list of baselines in Table 2, we had to rely on the reported numbers (as we could not reimplement and test every single one of these baselines). That’s why we compared FitVid with previous work on BAIR using FVD as it was the most commonly reported number. Simply put, we do not have the values for other metrics for other baselines in Table 2. Please note that reporting only FVD is quite common in video prediction literature. For example, check Table 1 of “Scaling autoregressive video models” or Table 1 and 2 of “CCVS: Context-aware Controllable Video Synthesis”.
>
> > Their qualitative results on the website look very similar to baselines (GHVAE) for me.
>
> Notable differences in the prediction quality is mostly visible in the details on the videos. For example, in RoboNet the differences are much more clear in the preserved and predicted details of the *pushed objects* (which is crucial for planning). Or in Human3.6M generated human motion looks much more natural compared to GHVAE.
>
> > Claim on Sec 5 about generalizing to unseen frames. It is a weird claim about video prediction as the unseen future itself is output. The model copy-pasting something from train set does not indicate generalization. I would just call it overfitting/memorizing train set. (okay w claim 2 but not claim 1)
>
> Thank you for mentioning this as there are a few nuances here. What we are claiming in (1) is that the model can still generate videos when conditioned on previously unseen frames. In Human3.6M this means that the model can still generate videos of humans doing things, when the input frame is a new subject (with new clothes, new pose etc). In this case, the model cannot simply copy/paste data from the training set because the training data set does not contain the new subject. However, the model manages to *generalize* to the new subject and recognize it as a human, and then move the new subject around, rather than generating noise or broken videos. In fact, what actually happens is that the model morphs the new subject into a human seen during training and then plays the rest from the memory while preserving the pose.  We updated the text to make this more clear.
>
> > Is the figure 6 results with or without data augmentation? If it’s without, will data augmentation alleviate overfit? If it is with data augmentation, direct comparison (either quantitatively or qualitatively) is expected.
>
> The results in Figure 6 are with augmentation. Without augmentation FitVid overfits on the data (Figure 7a), resulting in worse prediction quality:
>
> |                      | FVD   | PSNR | SSIM | LPIPS |
> |----------------------|-------|------|------|-------|
> | With Augmentation    | 385.9 | 27.1 | 95.1 | 0.026 |
> | Without Augmentation |  2224 | 25.8 | 92.2 | 0.063 |
>
> We will include this table in the appendix of the paper.

---

### Official Review · Reviewer_jPAY · 2021-11-07

**Correctness:** 3
**Technical Novelty And Significance:** 2
**Empirical Novelty And Significance:** 2
**Recommendation:** 5
**Confidence:** 3

**Main Review:**

(1). The paper tries to argue that previous papers suffer from underfitting are mainly a result of inefficient usage of model parameters. It may provide new insight for researchers to design models for video prediction.

(2). The writing of the proposed model is clear and easy to follow.

**Summary Of The Paper:**

This paper discusses a new framework named FITVID to handle the problem of video prediction. FitVid is built on existing modules like Sikp connection, Sequeeze and Excite, and LSTMs. FitVid only needs a simple training strategy and begins overfitting the datasets. Data augmentation is also adopted to handle the problem of overfitting.

**Summary Of The Review:**

However, the reviewer does have some concerns on the paper.
(1). A major concern of the paper is about the model's novelty. The reviewer has doubts on the argument that a new combination of existing techniques (BN, LSTM, S&E, skip connection) for the task of video prediction is significant enough to publish in ICLR. It will be important to show what is the design motivation of the combination and what it can outperform existing models on the task of video prediction.

(2). A comparison of components of different models is needed to help clarify the novelty of the proposed method. It will be also easier for readers to understand why previous models suffer from underfitting and how the model is more data efficient.

(3). Although the authors provide experimental results on 4 datasets. However, most of them contain only two simple baselines and the results on these datasets are not consistent. It will be more convincing if the authors can provide results on future and counterfactual predictions of the video dynamics on the CLEVRER[A] dataset and compare the proposed model' performance with recent stronger backbones like transformer [B], GNN[C] and Differentiable Physics Engine [D].

[A]. Yi K, Gan C, Li Y, et al. Clevrer: Collision events for video representation and reasoning[J]. arXiv preprint arXiv:1910.01442, 2019.
[B]. D. Ding, F. Hill, A. Santoro, and M. Botvinick. Object-based attention for spatio-temporal reasoning: Outperforming neuro-symbolic models with flexible distributed architectures. arXiv 2020.
[C]. Chen Z, Mao J, Wu J, et al. Grounding physical concepts of objects and events through dynamic visual reasoning[J]. arXiv 2021.
[D]. Ding M, Chen Z, Du T, et al. Dynamic Visual Reasoning by Learning Differentiable Physics Models from Video and Language[J]. arXiv  2021.

---

> ### Author Response · Authors · 2021-11-16
> **Authors response.**
>
> We would like to thank the reviewer for their time and detailed review.
> Regarding motivation, intuition and comparison with previous work, please read our aggregated response in
> https://openreview.net/forum?id=iim-R8xu0TG&noteId=WEQXp8Kw5Ta
>
> > The reviewer has doubts on the argument that a new combination of existing techniques (BN, LSTM, S&E, skip connection) for the task of video prediction is significant enough to publish in ICLR.
>
> We kindly disagree. Reducing the novelty of such paper to a “a new combination of existing techniques (BN, LSTM, S&E, skip connection)” is undermining the impact of a new architecture, considering the fact that many impactful papers of the past can be categorized as such. NVAE (NeurIPS 2020), EfficientNet (ICML 2019) and DC-GAN (ICLR 2015) to name a few examples of papers that their main contribution is a new architecture. However, all of these papers had a great impact on the field. This paper can be as impactful since it provides an easy-to-train, stable, state-of-the-art video prediction model that can be used for many downstream tasks as well as pushing the envelope on video prediction. This makes a good fit for a top tier conference such as ICLR.
>
> > A comparison of components of different models is needed to help clarify the novelty of the proposed method. It will also be easier for readers to understand why previous models suffer from underfitting and how the model is more data efficient.
>
> Please refer to our separate reply to this question which includes more intuition and details, comparison with previous work as well as new ablation studies.
> https://openreview.net/forum?id=iim-R8xu0TG&noteId=WEQXp8Kw5Ta
>
> > Although the authors provide experimental results on 4 datasets. However, most of them contain only two simple baselines and the results on these datasets are not consistent. It will be more convincing if the authors can provide results on future and counterfactual predictions of the video dynamics on the CLEVRER[A] dataset and compare the proposed model' performance with recent stronger backbones like transformer [B], GNN[C] and Differentiable Physics Engine [D].
>
> Our goal in selecting the baselines was to provide a comparison between FitVid and the state-of-the-art in video prediction. GHVAE (published in CVPR 2021) is the current state-of-the-art model in video prediction. We also provided comparisons with SVG (NeurIPS 2019) because it has the closest architecture compared to FitVid. And in Table.2, we compare the proposed method with even more models of all kinds, including more recent works such as VideoGPT (2021), CCSV (2021) and cINNS (2021)
> Please note that predicting video pixels is quite different from “reasoning on raw videos”. In the latter the goal is to learn good representation for down-stream tasks such as activity recognition or collision detection. However, in the former, the goal is to predict every single pixel of future video frames. Given this, we do not believe that comparing FitVid to CLEVRER is an appropriate comparison to do.

---

### Author Response · Authors · 2021-11-16
**Intuition and the rationale behind design decisions.**

We would like to thank the reviewers for their time and detailed reviews. A common comment in all the reviews was regarding the intuition behind the design decisions, as well as asking for a more detailed comparison with baseline models. In this reply, we provide a single response to these questions, while responding to the other questions separately. Our goal is to provide intuition behind out design decision as well as supporting these decisions with ablation studies. We updated the paper to include the new ablation results as well.

SVG, GHVAE, and FitVid all follow the same U-Net like architecture with skip connections and a dynamics model in the latent space. However, they are substantially different in their architectures and training details. SVG models the dynamics using ConvLSTMs and therefore maintains the spatial information of encoded frames in the dynamics model. SVG has to be trained using a learned prior and carefully adjusted beta in ELBO which can be hard to optimize. GHVAE breaks down the latent space into a hierarchical structure by sampling at every skip connection, which results in a larger model that cannot fit into memory and therefore it has to be trained at every hierarchy level in a greedy fashion.

In this paper, our goal was to find an architecture that can model the current video prediction benchmark and be easily trainable with no special tricks (Sec 3.2: What FitVid does *not* need). In summary:
1. Prior works address underfitting by making the model bigger, but this naive solution is terribly inefficient in terms of computation,training time and complexity. In FitVid our main goal was to find a more efficient and easy to train architecture *without* increasing the parameter count.
2. To achieve the goal in (1), we use non-hierarchical compressed latents (vs GHVAE) which provides a clear separation between frame encoders/decoders and dynamic models. This provides a separable design that can be broken down into independent pieces where each piece can be optimized separately.
3. Given the separation in (2), we replace the prior work encoder/decoder block with new blocks which are substantially more efficient in modeling the image frames. In order to show this efficiency, we ran another ablation study in which we replaced the encoder and decoder of SVG, one by one:

| Encoder | Decoder | FVD   | LPIPS |
|---------|---------|-------|-------|
| SVG     | SVG     | 155.2 | 0.026 |
| FitVid  | SVG     | 113.1 | 0.025 |
| SVG     | FitVid  |  76.9 | 0.020 |
| FitVid  | FitVid  |  **64.5** | **0.018** |

From this table, it can be seen that the new blocks are much more efficient at modeling the videos, resulting in improved prediction metrics, achieved when using FitVid encoder and decoder. Please note that the numbers reported for SVG are different from Table 1, because we used FitVid’s dynamic model in this study. We also searched for a sweet spot in terms of number of encoding and decoding blocks (after 1 million training steps):

| Encoder Blocks | Decoder Blocks | FVD   | LPIPS |
|----------------|----------------|-------|-------|
|              1 |              1 | 121.3 | 0.022 |
|              1 |              2 | 100.8 | 0.021 |
|              1 |              3 |  73.3 | 0.019 |
|              2 |              1 | 100.3 | 0.025 |
|              2 |              2 |  **64.5** | **0.018** |
|              2 |              3 | 176.6 | 0.028 |
|              3 |              1 |  88.8 | 0.019 |
|              3 |              2 |  83.9 | 0.019 |
|              3 |              3 |  78.6 | 0.018 |


This table clearly shows that it's not just about making the architecture bigger, but that we are actually benefiting from the particular design of our encoder and decoder i.e. having exactly 2 encoding blocks and 2 decoding blocks is the optimal spot for achieving the best results in a reasonable number of training iterations while maintaining the number of parameters in the same ballpark as previous work. While more encoding and decoding blocks add more parameters to the model, this table demonstrates that it does not necessarily lead to better performance.

4. Now with a more efficient architecture, we could optimize for an easier and more stable training procedure. In this work, we make training easier (1) removing the learned prior (which reduces the training complexity and provides more stability), (2) calculating the batch norm statistics across time vs batch (which allows for more stable training on small batch sizes, crucial for fitting into memory and avoiding training tricks such as greedy training), (3) changing the skip connections to residuals (which makes the model less sensitive the initialization) and (4) removing beta altogether, allowing for simply training using ELBO. The effect of these changes can be seen in ablation study at Table 4. We will include the new ablation studies in the appendix.

---

> ### Author Response · Authors · 2021-11-16
> **Further experiments**
>
> We hope that these additional ablations and comparisons help readers understand the importance of each of the individual design decisions in our architecture. No single choice by itself makes the model work great, but the particular combination of choices allow our proposed architecture to significantly outperform prior works. If you believe that these experiments still do not sufficiently address the question of which components of FitVid are important or how/why FitVid works, we would really appreciate some recommendations about which additional evaluations would better answer this question. We are happy to run additional experiments and evaluations to get to the bottom of this question, but at the same time we think it's important to recognize that, when it comes to research on improving neural network architectures, there is only so much that can be done to fully understand the precise reasons for improvements, and we believe our efforts are well in line (and indeed in excess of) comparable work on better neural network architectures. Thank you.

---

### Decision · Program_Chairs · 2022-01-20

**Decision:**

Reject

**Comment:**

This paper proposes a  variational video prediction model FitVid and attains a better fit to video prediction datasets.  The draft was reviewed by four experts in the field and received mixed scores (1 borderline accept, 3 reject). The reviewers raised their concerns on lack of novelty, unconvincing experiment, and the presentation of this paper. For a video prediction model, fitting a dataset is quite important. But AC agrees with the reviewer jPAY. It will be more exciting to build a causal model of the world and enable it to perform future and counterfactual prediction (e.g,  CLEVRER).  The authors are encouraged to consider the reviewers' comments when revising the paper for submission elsewhere.